# Electrically controlled superconductor-to-failed insulator transition and giant anomalous Hall effect in kagome metal CsV$_3$Sb$_5$ nanoflakes

Guolin Zheng[1,2,9], Cheng Tan[1,3,9], Zheng Chen[2,9], Maoyuan Wang[4,5], Xiangde Zhu[2], Sultan Albarakati[1], Meri Algarni[1], James Partridge[6], Lawrence Farrar[1], Jianhui Zhou [2] ✉, Wei Ning [2] ✉, Mingliang Tian [2,7] ✉, Michael S. Fuhrer [8] & Lan Wang [1,3] ✉

The electronic correlations (e.g. unconventional superconductivity (SC), chiral charge order and nematic order) and giant anomalous Hall effect (AHE) in topological kagome metals AV$_3$Sb$_5$ (A = K, Rb, and Cs) have attracted great interest. Electrical control of those correlated electronic states and AHE allows us to resolve their own nature and origin and to discover new quantum phenomena. Here, we show that electrically controlled proton intercalation has significant impacts on striking quantum phenomena in CsV$_3$Sb$_5$ nanodevices mainly through inducing disorders in thinner nanoflakes and carrier density modulation in thicker ones. Specifically, in disordered thin nanoflakes (below 25 nm), we achieve a quantum phase transition from a superconductor to a "failed insulator" with a large saturated sheet resistance for T → 0 K. Meanwhile, the carrier density modulation in thicker nanoflakes shifts the Fermi level across the charge density wave (CDW) gap and gives rise to an extrinsic-intrinsic transition of AHE. With the first-principles calculations, the extrinsic skew scattering of holes in the nearly flat bands with finite Berry curvature by multiple impurities would account for the giant AHE. Our work uncovers a distinct disorder-driven bosonic superconductor-insulator transition (SIT), outlines a global picture of the giant AHE and reveals its correlation with the unconventional CDW in the AV$_3$Sb$_5$ family.

The layered kagome metals AV$_3$Sb$_5$ (A = K, Rb and Cs) that possess topological electron bands and geometrical frustration of vanadium lattices are of great interests[1–3]. This is in no small part due to the many quantum phenomena that they support including unconventional SC[4–11], novel nematic order[12], chiral charge density order[13–22], giant anomalous Hall effect[23–25] as well as the interplay between two-gap SC and CDW in CsV$_3$Sb$_5$[26]. The unique coexistence of electronic correlations and band topology in AV$_3$Sb$_5$ allows for investigating intriguing

transitions of these correlated states, such as the superconductor-insulator transition (SIT), a protocol quantum phase transition (QPT) that is usually tuned by disorders, magnetic fields and electric gating[27,28]. Moreover, the origin of giant AHE in AV$_3$Sb$_5$ and its correlation with chiral CDW remain elusive[29,30], in spite of several recently proposed mechanisms including the extrinsic skew scattering of Dirac quasiparticles with frustrated magnetic sublattice[23], the orbital currents of novel chiral charge order[13] or the chiral flux phase in the CDW

phase[31]. Thus the ability to tune the carrier density and the corresponding Fermi surfaces would play a vital role in understanding and manipulating these novel quantum states and further realizing exotic QPTs.

In this work, we find that electrically controlled proton intercalation[32,33] exhibits crucial impacts on the superconducting state, CDW state and the associated AHE in $CsV_3Sb_5$ nanoflakes via disorders and carrier density modulation. In thinner nanoflakes (below 25 nm) with large gate voltages (e.g. with an amplitude above 15 V), the enhanced disorders from intercalated protons suppressed both CDW and superconducting phase coherence and gave rise to a SIT associated with the localized Cooper pairs, featuring a saturated sheet resistance reaching up to $10^6$ $\Omega$ for $T \rightarrow 0$, dubbed a "failed insulator". While in thicker $CsV_3Sb_5$ nanoflakes with much lower gate voltages (within 7 V), the superconducting transition instead retained with nearly unchanged sheet resistance in normal state at 5 K, indicating very limited impact of disorder. However, the Hall measurements demonstrate a large modulation of carrier density (with the modulation up to $10^{22}$ cm$^{-3}$) and the relevant Fermi surface topology changes from a hole pocket to an electron pocket. Consequently, we find that the giant anomalous Hall conductivity (AHC) with a maximal amplitude exceeding $10^4$ $\Omega^{-1}$ cm$^{-1}$ mainly pinned down to a narrow hole-carrier-density window around $p = (2.5 \pm 1.2) \times 10^{22}$ cm$^{-3}$ at low temperatures. Meanwhile, the AHE exhibits a clear extrinsic-intrinsic transition as the Fermi level shifts across the CDW gap near the saddle point. The observed giant AHE can be ascribed to the extrinsic skew scattering of the holes in the flat bands with nonzero Berry curvature by V vacancies and magnetic field tilted paramagnetic (PM) impurities.

## Results

The layered material $CsV_3Sb_5$ has a hexagonal crystal structure with space group P6/mmm (No. 191). As shown in the upper panel of Fig. 1a, the $V_3Sb$ layers are sandwiched by antimonene layers and Cs layers. X-ray diffraction (XRD) (see Supplementary Fig. 1), reveals a sharp (001) diffraction peak, indicating a single crystal possessing (001) preferred orientation, in line with previous work[4]. The striking feature of $CsV_3Sb_5$ is that the V atoms form a 2D kagome network. This frustrated magnetic sublattice of V was expected to induce novel correlation effects, such as spin-liquid states[34,35]. The lower panel of Fig. 1a illustrates a schematic of the gating device. A $CsV_3Sb_5$ nanoflake is mounted on the solid proton conductor with an underlying Pt electrode to form a solid proton field effect transistor (SP-FET). Next we

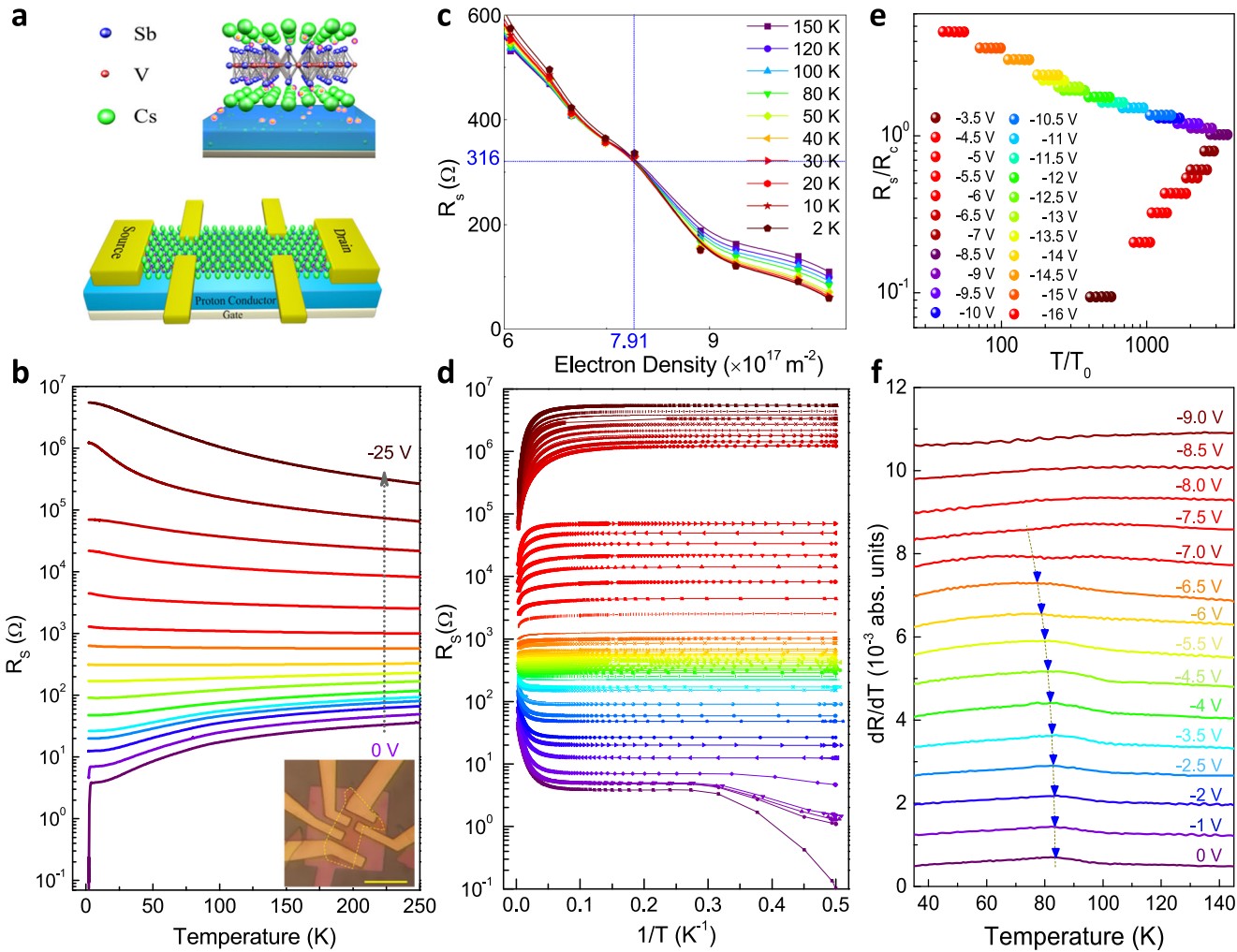

**Fig. 1 | Temperature-dependent longitudinal resistance curves under various gating voltages in device #5 (21 nm). a** Schematic of proton gating on $CsV_3Sb_5$ nanoflakes (upper) and Hall-bar device (lower). **b** Temperature dependence of sheet resistance at various gate voltages in device #5. **c** Sheet resistance as a function of charge density $n_e$ near SIT. The critical resistance $R_c \sim 316$ $\Omega$ is obtained with carrier density $n_c \sim 7.91 \times 10^{17}$ m$^{-2}$. **d** Sheet resistance as the function of 1/T under the same voltages. It exhibits a "failed insulator" with a large saturated resistance for $T \rightarrow 0$ K on the insulating side. **e** Multiple sets of $R_s$ (T, B) curves can collapse onto a single function, akin to a 2D SIT. **f** shows the derivatives of the resistance curves $R_{xx}$ (T) under different voltages. As the voltage changes from 0 V to $-6.5$ V, the CDW transition temperature $T_{cdw}$ gradually decreases from 85 K to 73 K at $V_g = -6.5$ V. CDW is largely suppressed when the gate voltage exceeds $-7.0$ V.

demonstrate how the proton intercalation significantly affects the SC state, CDW state and the giant AHE in CsV₃Sb₅ nanoflakes with distinct thicknesses.

## Protonic gate on thinner CsV₃Sb₅ nanoflakes

We first investigate the impacts of proton intercalation on the correlated electron states including SC and CDW in thinner CsV₃Sb₅ flakes below 25 nm. Figure 1b shows the temperature-dependent sheet resistance of device #5 with thickness of 21 nm at various gate voltages. A clear SC phase appears with the offset transition temperature $T_c^{offset}$ around 3 K in the absence of a protonic gate. Besides, a resistance anomaly as a characteristic of CDW near 80 K can be identified on $R_s -$ T curve (around 90 K in bulk, also in Supplementary Fig. 2). Applying a protonic gate, SC is clearly suppressed and disappeared when $V_g \leq -$ 2 V and $R_s$ fattens at around $V_g = -$ 11 V. At $V_g < -$ 11 V, temperature-dependent $R_s$ gradually exhibits an upturn at low temperature region and it eventually reaches up to above $10^6$ Ω at $V_g = -$ 25 V, indicating a quantum phase transition from a superconductor to an insulator. We plot the sheet resistance as a function of carrier density near SIT at temperatures between 2 K and 50 K and obtain a critical resistance $R_c \approx$ 316 Ω with a critical carrier density $n_c \approx 7.91 \times 10^{17}$ m⁻², as shown in Fig. 1c. Converting this critical resistance $R_c$ to the sheet resistance per layer, $R_{c/layer} = (\rho_c/l)$ with l being the thickness of a single layer, we get $R_{c/layer} = 7268$ Ω, very close to the quantum resistance of Cooper pair $R_Q \sim 6450$ Ω. In spite of the big $R_s$ reaching up to $10^6$ Ω on insulating side, however, a saturated resistance trend appeared for T → 0 K, as shown in Fig. 1d. Note that this insulating state with a saturated resistance for T → 0 K is not a typical insulator but a "failed insulator", probably due to the incoherent tunneling between localized Cooper pairs[36,37]. This type of superconductor to "failed insulator" transition has also been observed in another sample #8 in Supplementary Fig. 5 and mainly results from the enhanced effective disorders due to the intercalated protons in the thinner nanoflakes with higher gate voltages[38].

SIT can be usually characterized by two distinct scenarios (bosonic and fermionic) according to the nature of insulating phase, while the finite-size scaling analysis could yield its critical exponents and further reveal the universality class of QPT[39–42]. To get the critical exponents of SIT, we plot more than twenty sets of $R_s(T,B)$ curves and find that they can collapse onto a single function, as predicted for a 2D SIT. The appearance of flatten resistance near $R_c$ suggests the bosonic nature of SIT, in which the coherent Copper pairs in the SC phase are localized by disorders with loss of macroscopic phase coherence in the insulating phase[27,28]. The finite size scaling dependence of $R_s$ on T and a tuning parameter has the form $\rho(T,n_s) = \rho[\frac{T}{T_0(n_s)}]$ with $T_0 \propto |n_s - n_c|^{vz}$ where $n_s$ is the charge density, $n_c$ is the critical carrier density with the value of $n_c \approx 7.91 \times 10^{17} m^{-2}$ (or $1.3 \times 10^{18}$ m⁻² in device #8) and $T_0$ is the scaling parameter which approaches to zero at $n_s = n_c$. $v$ is the correlation-length exponent and z is the temporal critical exponent[42]. By extracting the exponent product $vz$ and plotting $\ln T_0$ versus $\ln|n_s - n_c|$ curve, we can obtain $vz = 1.85$ (or 1.68 in device #8) with an uncertainty of ±0.14 (see Supplementary Fig. 3). This estimated exponent product is close to that of magnetic-field tuned SIT in the hybrid system of SC indium islands deposited on 2D indium oxide thin film[43], which was also attributed to localization of persisting Cooper pairs. Note that the critical exponent here is distinct from those of 2D conventional models for SIT[28] such as classical percolation model ($vz = 4/3$), quantum percolation model ($vz = 7/3$), which probably results from the complexity of SC gap and multiple impurities[44]. Thus, CsV₃Sb₅ provides us with a unique example system to explore rich QPTs involving intrinsic superconductors with topological energy bands and frustrated kagome lattice.

Figure 1d shows the derivatives of the RT curves at various $V_g$. Interestingly, CDW transition temperature $T_{CDW} = 85$ K at $V_g = 0$ V gradually decreases to 73 K at $V_g = -$ 6.5 V where SC has been

suppressed. More importantly, at $V_g \leq -$ 7 V, we found this resistance anomaly totally disappeared on RT curve, indicating the disappearance of the CDW. The non-synchronous disappearance of SC and CDW reveals that SC is more sensitive to disorders. The suppression of CDW is also consistent with recent works for CsV₃Sb₅ under high pressures, possibly due to band reconstructions or Fermi level shift[4–8]. It is clear that the protonic gate significantly modifies the CDW and SC phases, facilitating the further investigation of the intertwinement among these novel electronic correlations in AV₃Sb₅.

## Protonic gate on thicker CsV₃Sb₅ nanoflakes

For a given gate voltage, the thick samples would diminish the impact of disorder of intercalated protons, leaving a large tune of the carrier density. Let us concentrate on the significance of the carrier density modulation on the AHE in CDW phase. We choose thicker CsV₃Sb₅ nanoflakes with much lower gate voltages within 7 V and find that the proton intercalation mainly changes the carrier density in those thicker ones, leading to only a slight modulation of SC transition temperature (Supplementary Fig. 6), which is consistent with recent reports[45–47]. Figure 2a shows the Hall traces of device #4 (around 80 nm) at various temperatures and selected gate voltages. At low magnetic fields, the Hall resistance $R_{yx}$ at $V_g = 6.4$ V exhibited a non-linear behavior at 5 K. This antisymmetric "S"-shape $R_{yx}$ was attributed to field induced AHE in KV₃Sb₅[23] and CsV₃Sb₅[24]. At high fields, $R_{yx}$ exhibits an approximately linear field dependence associated with the ordinary Hall effect induced by the Lorentz force. Note that AV₃Sb₅ is a multi-band kagome metal with its transport properties mainly determined by the hole pocket near the M points[48]. Thus we can use a simple band model to fit this linear Hall resistivity at high field region and extract the approximate hole carrier density at the M-points. When $6.4V \geq V_g \geq -$ 2.7 V, the Hall traces in device #4 exhibit two distinct features. For each gating voltage, the temperature-dependent Hall effects demonstrate a sign reversal at the critical temperature T*, probably due to the temperature-induced band renormalization[48]. In addition, the Hall slope decreases gradually as the voltage is swept towards −2.7 V, indicating a gradual increase of the hole carrier density. At $V_g = -$ 4.6 V, however, the Fermi surface topology suddenly changes from a hole pocket to an electron pocket with a negative Hall slope. This doping-induced sign reversal of Hall resistance has also been observed in other samples (Supplementary Fig. 9). In contrast to the hole pockets, the Hall traces in the electron pockets exhibit no sign reversal as the temperature is increased, as shown in the bottom right panel of Fig. 2a at $V_g = -$ 6.1 V, indicating a dramatic suppression of T* in the electron pocket. We further plot the gate-dependent carrier density at 5 K in device #4 in Fig. 2b. At $V_g = -$ 4.6 V, the Fermi level is shifted across the electro-hole crossover point. We note that the discontinuity of the carrier density under gate voltage probably stems from the complex evolution of density of states (DOS) during the proton intercalation (See theoretical calculations below). Figure 2c shows the carrier-density (obtained at 5 K) dependent T* for bulk crystals and four nanodevices. In hole pockets, a higher hole density may lead to a smaller T*. However, T* approaches 0 K for the electron pockets, due to the sudden change of the Fermi surface topology.

We now discuss the gate-dependent AHE in CsV₃Sb₅. The total Hall resistivity $\rho_{yx}$ consists of two components[49]: $\rho_{yx} = \rho_{yx}^N + \rho_{yx}^A$, with $\rho_{yx}^N$ the normal Hall resistivity and $\rho_{yx}^A$ the anomalous Hall resistivity. In order to extract the AHE component, the Hall resistivity was linearly fitted at high field to subtract $\rho_{yx}^N$. Figure 3a shows the gate-dependent anomalous Hall resistivity $\rho_{yx}^A$ of device #4 at 5 K under various gate voltages. The maximum $\rho_{yx}^A$ occurs at $V_g = 4.5$ V with an amplitude of 0.041 μΩ•cm that is approximately eight-fold larger than the minimum $\rho_{yx}^A$ (0.0048 μΩ•cm) measured at −4.6 V. Interestingly, the AHE also exhibits a sign reversal at $V_g = -4.6$ V which is probably due to the sign change of the Berry curvature in different energy bands, as shown in

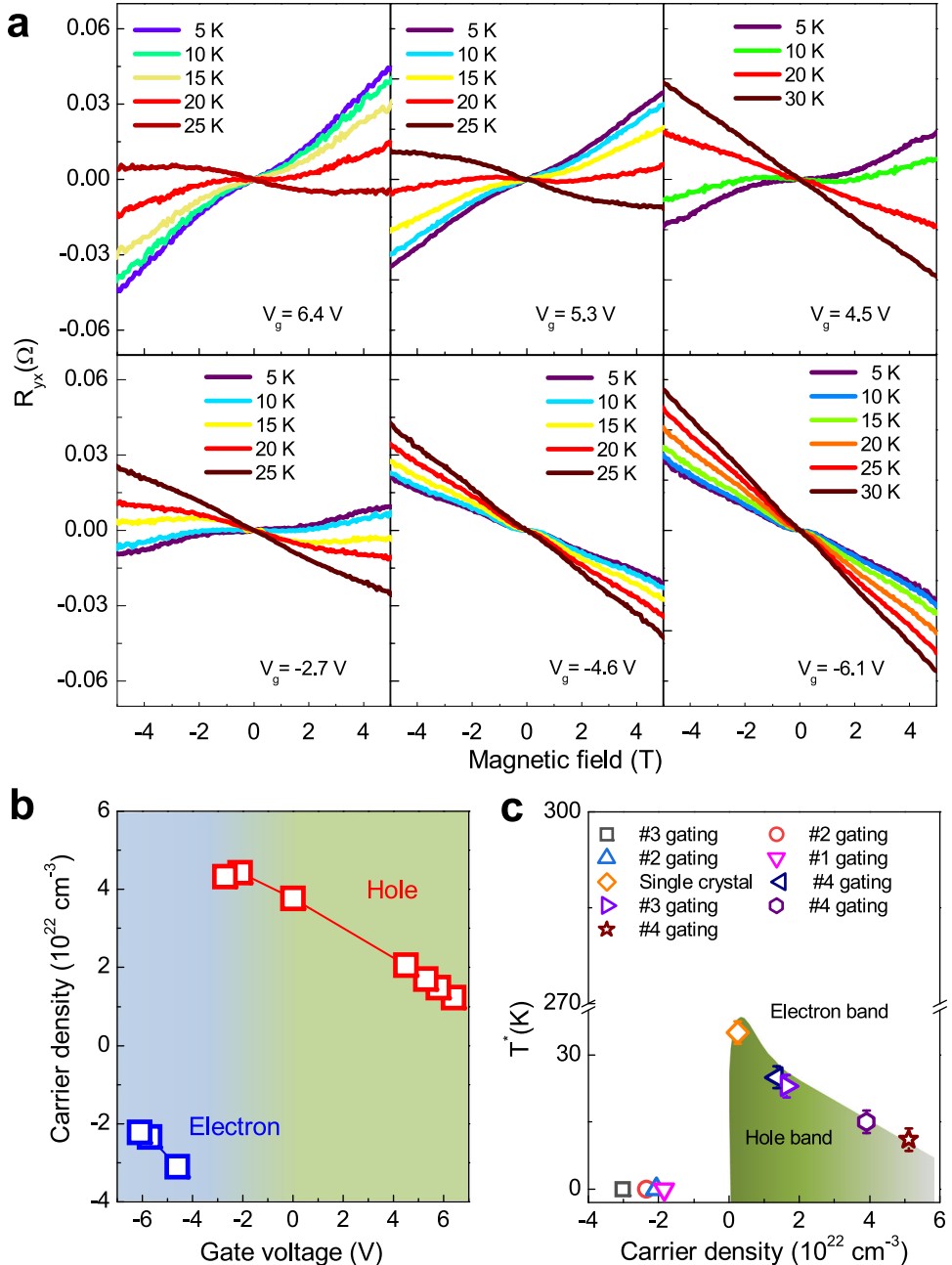

**Fig. 2 | Gate-tuned Hall resistance and carrier density dependent band topology. a** Temperature-dependent Hall effect in device #4 under different gating voltages. **b** Gate-dependent carrier density in device #4 at 5 K. Sweeping the gate voltage from 6.4 V to −6.1 V, the band structure evolves from a hole band to an electron band in the low temperature region. **c** Carrier density dependent T* in different samples.

Fig. 3a. To get the AHC $\sigma_{xy}^A$, we first convert the Hall resistivity into the Hall conductivity $\sigma_{xy} = \rho_{yx}/\left(\rho_{yx}^2 + \rho_{xx}^2\right)$, followed by linearly fitting the conductivity at high field and subtracting the normal Hall conductivity $\sigma_{xy}^N$. Figure 3b displays the non-monotonic variation of both the AHC and the anomalous Hall angle (AHA) $\theta = |\sigma_{xy}^A/\sigma_{xx}|$. The maximal AHC reaches $1.24 \times 10^4\,\Omega^{-1}\mathrm{cm}^{-1}$ with an AHA of 2.2% at 4.5 V. Moreover, the AHC (AHA) can be modulated by more than ten times in device #4, revealing the high tunability of the AHE in CsV$_3$Sb$_5$. Figure 3c shows two carrier density regions that exhibit large AHE for different devices. The first region is mainly pinned down in the hole pocket around p = $(2.5 \pm 1.2) \times 10^{22}\,\mathrm{cm}^{-3}$ with the maximum AHC exceeding $10^4\,\Omega^{-1}\mathrm{cm}^{-1}$. Remarkably, another large AHE appears in the electron pocket between n = $(3 \pm 0.6) \times 10^{22}\,\mathrm{cm}^{-3}$ with the AHC around $5 \times 10^3\,\Omega^{-1}\mathrm{cm}^{-1}$. Shifting away from these two regions, however, the AHC either keeps a

finite value or approaches zero for devices #1 and #2 (Supplementary Fig. 9).

The scaling law between AHC and $\sigma_{xx}$ may assist in identifying the underlying mechanism of the AHE[49,50]. Figure 4a displays the scaling relations $\sigma_{xy}^A$ vs $\sigma_{xx}$ at various gate voltages and temperatures. In the high conductivity region ($\sigma_{xx}$ exceeds $5 \times 10^5\,\Omega^{-1}\mathrm{cm}^{-1}$), the maximal AHC obtained in device #4 (4.5 V) and device #7 (at 3.8 V) can be well captured by a linear scaling relation $\sigma_{xy}^A \propto 0.14\,\sigma_{xx}$, revealing that the skew-scattering mechanism may dominate the AHE[49–53]. The possible side jump contribution is also discussed in Supplementary section 9. However, at $V_g = -4.6$ V, a finite AHC around $10^3\,\Omega^{-1}\mathrm{cm}^{-1}$ is approximately independent of the longitudinal conductivity $\sigma_{xx}$, implying that the intrinsic AHE from the Berry curvature becomes dominant. For other gating voltages, AHEs are likely linked to the mixing region. The

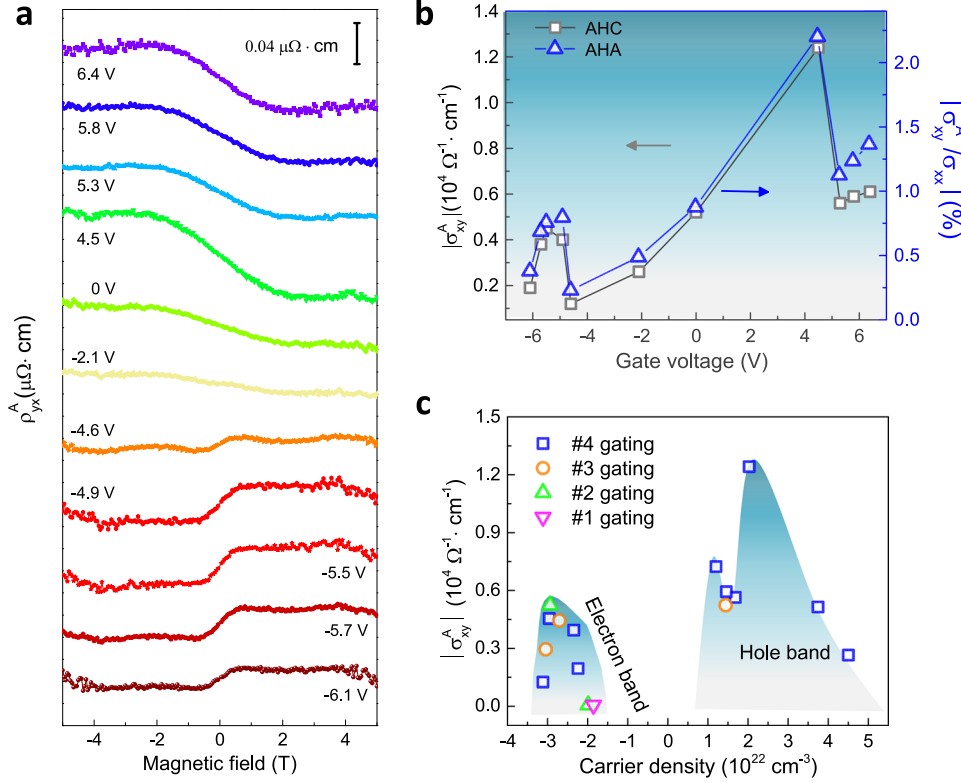

**Fig. 3 | Gate-tuned giant anomalous Hall effects in device #4. a** Gate-dependent anomalous Hall effect at 5 K after subtraction of the linear Hall background in the high field regions (ordinary Hall part). **b** Gate-dependent AHC and anomalous Hall angles (AHA). **c** Carrier density dependent AHC in different devices #1, #2, #3 and #4. The maximum AHC occurred with a hole carrier density of -2 × 10²² cm⁻³.

gate-induced crossover between the extrinsic (at $V_g = 4.5$ V) and intrinsic regimes (at $V_g = -4.6$ V) reveals a strong dependence of AHE on the Fermi energy of $CsV_3Sb_5$.

To gain more insights into AHE, we performed the first-principles calculations of the band structure, the DOS and the intrinsic AHC (Supplementary Fig. 12). The calculated AHC due to the field induced magnetization of the spin of V atoms over a broad energy region, exhibits a maximum ($-1500 \Omega^{-1} cm^{-1}$) in the hole band, one order smaller than the maximum experimental value. This suggests that the intrinsic contribution from the Berry curvature of single-particle energy bands should not dominate the giant AHE in experiments. Note that, because of the tiny observed magnetic moments of V atoms[45], the realistic intrinsic AHC from field-induced magnetization should be much smaller than the observed intrinsic AHC. It has been known that the extrinsic skew scattering of AHE essentially originates from the asymmetric scattering of carriers by nonmagnetic/magnetic impurities. Usually, there are three distinct scenarios to produce the extrinsic skew scattering and the resultant AHE[49]. By careful examination of each scenario, we could exclude the Kondo scattering and resonant skew scattering (Supplementary section 9). We found that the scenario associated with finite Berry curvature of energy bands and scattering by nonmagnetic/magnetic impurities primarily accounts for extrinsic AHE in $AV_3Sb_5$[49,54].

## Discussion

We further investigate the impact of charge doping on the band structure and AHE. Since the charge doping in $CsV_3Sb_5$ is orbitally selective, the hole (electron) doping can significantly shift the van Hove singularity (VHS1) upward (downward) with respect to the Fermi level within the rigid-band approximation, as shown in Fig. 4b. In our pristine $CsV_3Sb_5$ single crystal, the Fermi level lies slightly above VHS1 near the M point (Supplementary section 2), with some nearly flat bands consist of $d_{xz,yz}$ and $d_{xy,x^2-y^2}$ orbitals of V atoms (Supplementary Fig. 13)[55–58].

When $T < T_{cdw}$, a CDW gap opens near VHS1, splits the bands at VHS1 into two sub-bands and suppresses the DOS near the Fermi level, as shown in Fig. 4c. Accordingly, the Fermi level in bulk $CsV_3Sb_5$ lies in the CDW gap[11,55] near the M point (red dashed arrow), exhibiting a large AHE. In exfoliated $CsV_3Sb_5$ nanoflakes, the Fermi level approaches the lower sub-band due to the increasing Cs vacancies and the AHC at $V_g = 0$ V reduces to about one third of the maximal value, i.e., 4500 $\Omega^{-1}$ cm⁻¹. Applying a negative voltage will accordingly lower the Fermi level (details in Supplementary section 14) and generate a relatively large AHE region (with AHC around 5000 $\Omega^{-1}$ cm⁻¹) in the electron pockets. At $V_g > 0$ V, however, the Fermi level will be shifted upward and back to the upper sub-band again (dashed red arrow), the giant AHE reappears at 4.5 V in device #4. This giant AHE primarily comes from the skew scatterings of holes in the nearly flat bands at VHS1 with nonzero Berry curvature by the V vacancies and/or PM impurities. The large discrepancy of AHE in the electron and hole pockets is consistent with asymmetric distribution of DOS in the CDW bands near the M points[11]. Note that the intrinsic AHC at $V_g = -4.6$ V near the electron-hole crossover point is mainly ascribed to the large suppression of the DOS at the middle of the CDW gap. After evaluating the possible AHE from the electron near the Γ point and the Dirac bands, we find this intrinsic AHE mainly originates from the recent chiral charge order forming from the electronic states near the saddle point[59–61].

In summary, we revealed two major impacts of proton intercalation on $CsV_3Sb_5$, inducing disorders in thinner nanoflakes and carrier density modulation in thicker ones. In thin nanoflakes below 25 nm with $|V_g| \geq 15$ V, we first observed a distinct superconductor-to-"failed insulator" transition associated with localized Cooper pairs. In thicker nanoflakes, a moderate gate voltage can lead to a large modification of the carrier density and induce a clear extrinsic-intrinsic transition of AHE. The giant AHE in $AV_3Sb_5$ can be attributed to the intense extrinsic skew scattering of holes in the nearly flat bands with finite intrinsic AHE in the CDW phase at the saddle points by multiple impurities. This

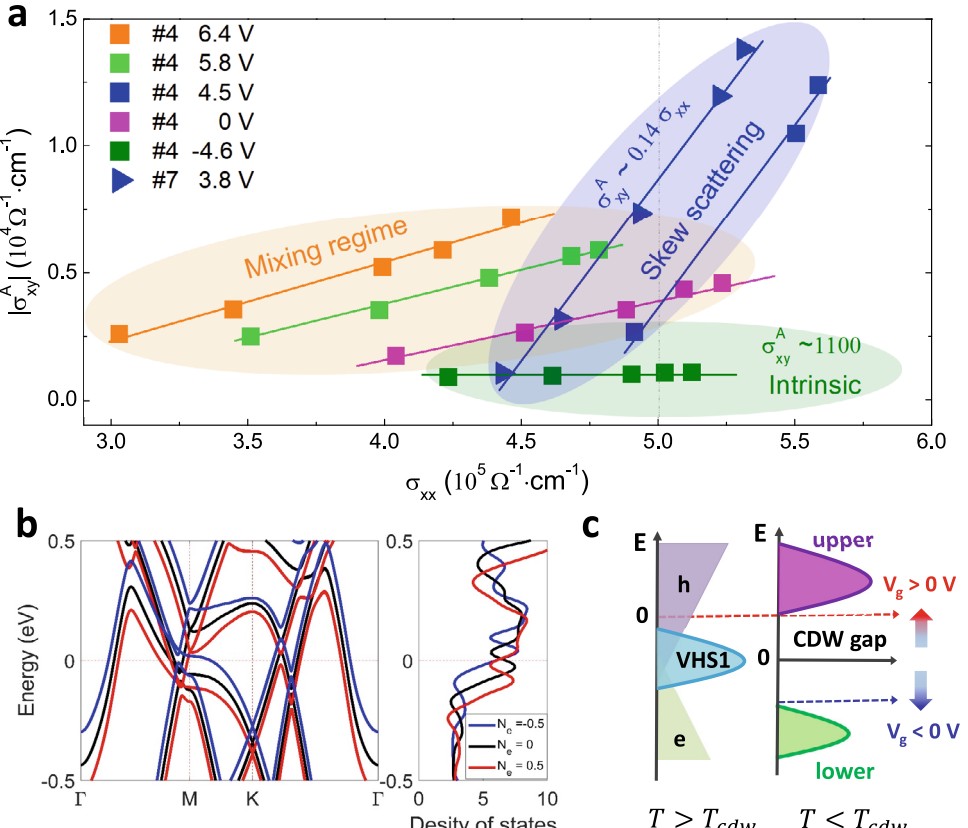

**Fig. 4 | Scaling relation, band structure and the density of states in gated CsV3Sb5. a** The scaling relation of AHC against the longitudinal conductivity in devices #4 and #7 (with thickness around 50 nm). Near the high conductivity region (above $5 \times 10^5$ $\Omega^{-1}$ cm$^{-1}$), the giant AHE is dominated by the skew scattering (#4 at 4.5 V and #7 at 3.8 V). At −4.6 V (#4, electron band), the AHE is dominated by the intrinsic Berry curvature. **b** Band structures of the paramagnetic phase with different doping levels in CsV$_3$Sb$_5$. Ne refers the charge number in each primitive cell.

**c** Illustration of the evolution of Fermi energy under different gate voltages. Red dashed arrow shows the probable Fermi level in bulk crystal (slightly above VHS1). Cs vacancies in CsV$_3$Sb$_5$ nanoflakes will significantly lower the Fermi level (black arrow). Applying a negative (positive) voltage will shift the Fermi level downward (upward). The giant AHE occurs when the Fermi level approaches the upper subband with relatively large DOS.

significant and electrically controlled SIT and AHE in CsV$_3$Sb$_5$ should inspire more investigations of the relevant intriguing physics and promising energy-saving nanoelectronic devices.

## Methods

### Single crystal growth

Single crystals of CsV$_3$Sb$_5$ were synthesized via Sb flux method. The elemental Cs, V and Sb were mixed at a molar ratio of 1:3:20, and loaded in into a MgO crucible. This process was performed in a glove box in Ar ambience. Then the crucible was sealed in a vacuumed quartz tube. The ampule was slowly heated to 1000 ˚C and kept for 20 h. After cooling at a rate of 2 °C/min, the extra flux was removed by fast centrifuging at 640 °C. (Also in Supplementary Section 1).

### Device fabrication and transport measurements

Solid protonic electrolyte was prepared by the sol-gel processes by mixing tetraethyl orthosilicate (from Alfa Aesar), ethanol, deionized water, phosphoric acid (from Alfa Aesar, 85% wt%) with a typical molar ratio 1:18:6:0.03. The mixed solution was then stirred for 2 h and annealed for another 2 h at 50 °C in a sealed bottle to form polymerized Si−O−Si chains. Finally, the substrate with bottom gate electrodes was spin-coated with the prepared protonic solution and baked at 150 °C for 25 mins. Transport measurements were performed in a commercial Physical Property Measurement System (PPMS) with magnetic field up to 9 T and a commercial magnetic property measurement system (MPMS) with magnetic field of 7 T.

## Data availability

The data used in Figs. 1–4 of the main text are provided in the Source Data. Additional data related to this study are available from the corresponding authors upon reasonable request. Source data are provided with this paper.

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

## Acknowledgements

The authors thank Y. M. Dai, H. LaBollita, Y. M. Li, H. W. Liu, K. Nakayama, Q. Niu, T. Sato, J. M. Tranquada, C. Xiao, Z. Y. Wang, H. Yang, J. J. Ying and L. Yu for insightful discussions. This research was supported by the Australian Research Council Centre of Excellence in Future Low-Energy Electronics Technologies (Project No. CE170100039), National Key R&D Program of the MOST of China (Grant No. 2022YFA1602603), the National Natural Science Foundation of China (Grants No. 12274413, U19A2093, U2032164, 12174394), the HFIPS Director's Fund (Grant No. YZJJQY202304) and the CASHIPS Director's Fund (Grant No. E26MMG71131). This work was also partially supported by Youth Innovation Promotion Association of CAS (Grant No. 2021117) and the High Magnetic Field Laboratory of Anhui Province.

## Author contributions

L.W. and M.T. conceived the project. G.Z. and C.T. fabricated the devices and performed the transport measurements, assisted by S.A., M.A. and L.F., Z.C., X.Z. and W.N. synthesized the single crystals. M.,W. and J.Z. provided theoretical support. G.Z., C.T., J.Z., J.P., M.S.F., M.T. and L.W. analyzed the data and wrote the manuscript with assistance from all authors.

## Competing interests

The authors declare no competing interests.

## Additional information

[1]ARC Centre of Excellence in Future Low-Energy Electronics Technologies (FLEET), School of Science, RMIT University, Melbourne, VIC 3001, Australia. [2]Anhui Province Key Laboratory of Condensed Matter Physics at Extreme Conditions, High Magnetic Field Laboratory, HFIPS, Chinese Academy of Sciences (CAS), Hefei 230031 Anhui, China. [3]Department of Physics, and Lab of 2D Materials and Quantum Devices, School of Physics, Hefei University of Technology, Hefei, Anhui 230009, China. [4]Department of Physics, Xiamen University, Xiamen, Fujian, 361005, China. [5]International Center for Quantum Materials, School of Physics, Peking University, Beijing 100871, China. [6]School of Science, RMIT University, Melbourne, VIC 3001, Australia. [7]School of Physics and Optoelectronic Engineering, Anhui University, Hefei 230601 Anhui, China. [8]ARC Centre of Excellence in Future Low-Energy Electronics Technologies (FLEET), Monash University, Melbourne, VIC 3800, Australia. [9]These authors contributed equally: Guolin Zheng, Cheng Tan, Zheng Chen. ✉e-mail: jhzhou@hmfl.ac.cn; ningwei@hmfl.ac.cn; tianml@hmfl.ac.cn; wanglan@hfut.edu.cn

