## [Peer Review File · Nature Communications]

REVIEWER COMMENTS

Reviewer #1 (Remarks to the Author):

The authors reported a detailed study on charge modulation in CsV₃Sb₅ flakes using a protonic gating. They found a superconducting-insulator transition and an interesting evolution of carrier density and AHE with the variation of gating voltage. They argued that the dramatic change of AHE with decreasing gating voltage is attributed to the extrinsic-to-intrinsic transition of AHE. This study is worthwhile but, as a theorist, I am not persuaded that this study will significantly advance our understanding exotic phenomena in kagome metals. The manuscript needs to clarify the following points,

1. With increasing of disorder, SIT is theoretically expected. Why is it exotic here? In addition, the authors only show the data for the device 5 with a thickness of 21 nm. How about the thinner flakes? Does the SIT only occur for thin flakes below a critical thickness?
2. The AHE is usually defined as the Hall effect without an external magnetic field. But here the Hall resistance always vanishes at a zero magnetic field. Why is it an AHE in the present experiments?
3. As the kagome metals CsV₃Sb₅ is a multi-orbital system, containing both holes and electrons, the carriers can exhibit complicated temperature dependent behavior. Therefore, a sign change of Hall resistance may not denote a Lifshitz transition. Using the protonic gating, the introduced disorder effect can also affect the carrier mobility. If the authors consider both electron and hole carriers, how will this affect the analysis for the evolution of Hall resistance?
4. Fig.2 only shows the data for thicker flakes (about 80 nm). Will similar behavior occur in thinner and thicker flakes? Is the sign reversal in Hall resistance a universal feature in kagome nano flakes?
5. In principle the gating method should introduce a continuous charge tuning. Why is the carrier density in Fig.3c discontinuous?
6. The kagome metals have been experimentally shown to be nonmagnetic but time-reversal-symmetry broken (by μ SR measurements Ref. 60,61). However, in the theoretical calculations, the authors used a ferromagnetic state to calculate the AHC with a relatively large magnetic moment $0.25 \mu_B$. The assumed ground state is not consistent with experiment data.

Reviewer #2 (Remarks to the Author):

In this manuscript, Zheng et al studied the SIT and the AHE in kagome metal CsV₃Sb₅ nanoflakes with various thicknesses by using a protonic gate. In a 21-nm thin flake device, they observed a direct SIT and a large suppression of CDW induced by gating, which were interpreted in terms of the phase fluctuation due to enhanced-disorder. Meanwhile, in thick devices, electric gating changes the carrier density and shifts the Fermi level position across the CDW gap and gives rise to giant AHE. By further providing first-principle calculations, the authors demonstrated the giant AHE is induced by the skew scattering mechanisms of holes in the flat bands with finite Berry curvature. This experiment stands in the continuation of efforts of the studies of the recently discovered kagome material AV₃Sb₅ that incorporates electron correlation effects, band topology and geometric frustration. Although there have already been relevant studies on the transport properties of AV₃Sb₅, the underlying physics of the SIT and the AHE have not been well addressed. Therefore, the electrical controls of SIT and AHE by gating have an opportunity to provide noteworthy results and may deepen people's understanding on these interesting topics.

I appreciate the authors' efforts in the manuscript. However, I find the current data does not support the conclusions. More reproducible experimental results are required to support the claims. Moreover, there may be a flaw in the AHE data analysis process, which makes the interpretation of the AHE questionable. In spite of these concerns, I still find experiments interesting. However, at the current stage, I cannot recommend this article for publication without significant revisions.

Please consider the following questions and comments:

1. It seems the SIT and the AHE part are two independent stories performed in samples with thin and thick thickness respectively. The connection between the two parts is too weak. To enhance the readability, I suggest the authors discuss the relationship between these two parts, rather than simply piecing them together.
2. Is the observed SIT reproducible? The authors only provide the data of one sample in the main text (device #5). The quantized resistance $h/4e^2$ at the transition point is regarded as an important signature of SIT. However, when looking at the "SIT" data of other samples in the supplementary, it is clear that the critical resistance is not $h/4e^2$. Even considering the different thickness, one cannot get the quantized value $h/4e^2$. Therefore, it is likely that the observed $RQ = h/4e^2$ in device #5 is accidental.
3. To convince general readers that the phase transition is universal, more data analysis is needed. What are the critical exponents of the two samples (12 and 14 nm devices) shown in the supplementary materials?
4. In previous studies of SIT, the sheet resistance R_S is defined as the resistance divided by area. In this case, the critical resistance at the superconductor to Bose insulator transition point reaches $RQ = h/4e^2$. However, in the current manuscript, R_S has a different definition. To get a quantized value, the number of layers is considered. This is more like a definition of resistivity rather than resistance. For the 2D SIT phenomenon, it is the 2D sheet resistance R_S rather than the RQ divided by layer number that should be universal. The authors should clarify this issue. Moreover, the protonic gating

ability and the induced transport responses vary a lot for different layer. It is inappropriate to directly divide R by layer number.

5. In Fig. 1b, there is a RS saturation at low temperatures. It is apparently different from a true insulating state that exhibits exponential diverging behaviors (see ref. 27). I suggest the authors plot the resistance as a function of $1/T$. In the community of SIT, people usually adopt this method to identify whether the phase transition is an SIT or an SMT. In addition, in AV3Sb5, there have already been reports on the metallic state rather than an insulator state when SC is killed (ref. 38). The authors should be careful when claiming a Bosonic SIT.

6. How to demonstrate the direct SIT is induced by enhanced disorders? What kind of disorder plays a role here?

7. The authors stated that the protonic gate plays different roles in thin and thick flakes, which are significant for the SIT and AHE. An important question is how the authors could distinguish between intercalation and carrier density modulation simply based on the current transport experiments?

8. An “S”-shaped R_{xy} was attributed to the field induced AHE (Fig. 2a). How to exclude the possibility that it is due to the coexistence of electron- and hole-type carriers?

9. A sign reversal was observed in the temperature dependent measurements and was attributed to the temperature-induced Lifshitz transition. Is it possible that this sign reversal is induced by the variation of gating ability at high temperatures?

10. When discussing the gate-dependent AHE, the authors decomposed the total Hall resistivity ρ_{-xy} into two parts: the normal Hall effect ρ_{-Nxy} and the anomalous Hall part ρ_{-Axy} . To quantitatively compare the AHE in AV3Sb5 with that in other systems, one has to convert the resistivity ρ_{-Axy} into conductivity σ_{-Axy} first. However, the way that the authors got σ_{-Axy} is wrong. The correct procedure is to transform ρ_{-xy} into σ_{-xy} through the tensor relationship $\sigma_{xy} = \rho_{xy} / (\rho_{2xy}^2 + \rho_{2-xx}^2)$ first, and then extract the σ_{-Axy} by subtracting the normal Hall conductivity σ_{-Nxy} (see refs. 23 and 47). However, in the current work, the authors first got ρ_{-Axy} , and then get σ_{-Axy} using the formula $\sigma_{Axy} = \rho_{Axy} / (\rho_{Axy}^2 + \rho_{2-xx}^2)$. This data analysis operation is invalid and may lead to serious problems. Because the giant AHE is one major selling point, the authors should carefully check the data analysis process and replot the AHC figures.

11. In Fig. 4a, the authors summarized the relation between σ_{Axy} and σ_{xx} , and claimed “the AHE in device #4 can be well captured by a linear scaling relation”. From the figures, I cannot see any linearity between σ_{Axy} and σ_{xx} . In my opinion, it is unscientific to do linear fitting only based on three data points.

12. In the AHE community, double log-scale plots are usually adopted when σ_{Axy} and σ_{xx} span several orders of magnitude (see ref. 24). Obviously, the variations of σ_{Axy} and σ_{xx} are too small here. It is unnecessary and misleading to use double-log scales.

13. The giant AHE was attributed to the skew scattering of holes by impurities. Could the authors estimate the level of magnetic impurities in the samples? For a nonmagnetic SC material, it is hard to imagine an extrinsic mechanism could give rise to such a large AHC.

Reviewer #3 (Remarks to the Author):

In manuscript “Electrically controlled superconductor-insulator transition and giant anomalous Hall effect in kagome metal CsV₃Sb₅ nanoflakes” by G. Zheng et al., they present the results of protonic gate control of CsV₃Sb₅ nanoflakes.

They succeeded in observing the superconductor to insulator transition. In addition, they demonstrated the tuning of anomalous Hall effect by the protonic gate.

They showed many results and each data looks quite interesting. However, it is difficult to understand the overall feature of this work from this manuscript. For example, Fig.4a seems to be quite interesting result because the anomalous Hall effect was able to be controlled into Skew scattering regime by gate. However, they plot only their present data on this Figure. By adding other data points from previous reports such as Ref[24 (PRB 104)], we can find the importance of this work. As for the superconducting issues, I recommend to depict the phase diagram T_c , T_{CDW} , T^* vs gate voltage or carrier density. With such phase diagram, we can understand smoothly what they achieved.

Regarding the analysis of Hall effect, I'm confused about the labels: R_{xy} and ρ_{xy} . I'm sure the equation: $\sigma_{xy} = -\rho_{xy} / (\rho_{xy}^2 + \rho_{xx}^2)$ in page.8. However, Hall resistance and Hall resistivity should be R_{yx} and ρ_{yx} . So that, the slope of Hall effect become negative at $V_g = 4.5V$ and 5 K, because the y-axis of Fig.2a is R_{xy} , not R_H ($R_{yx} (= -R_{xy})$). If they define the y-axis of Fig.2a is Hall resistance, the carrier type is correct as hole at $V_g=4.5V$ and 5 K, but the sign of anomalous Hall effect become negative. These relationship is opposite in Fig.3b. The authors should clarify this point.

Thus, at the moment, I would not recommend the publication of this work in Nat. Commun.

Re: NCOMMS-22-12877 “Electrically controlled superconductor-insulator transition and giant anomalous Hall effect in kagome metal CsV_3Sb_5 nanoflakes” by Guolin Zheng, Cheng Tan, Zheng Chen, et al.

We first thank the three reviewers for their evaluations and instructive suggestions, which indeed help us to largely improve our manuscript. The follows are our point-to-point responses to all of reviewers’ comments. The ‘reply’ is marked in blue color and ‘revision’ is in green color.

List of change

1. We fabricated more devices and reproduced the superconductor-to-failed insulator transition in a new device (device #8). Fig. 1 is updated accordingly in the revised version.
2. We further clarified the multi-band characteristic of AV_3Sb_5 and specified the method used to estimate the carrier density at the M point.
3. An unusual saturated resistance state with $T \rightarrow 0 \text{ K}$ has also been discussed in both devices #5 and #8, which is dubbed as a “failed insulator” state due to the incoherent tunneling between localized Cooper pairs.
4. The reason for the sign reversal of Hall resistance is further discussed, it is now attributed to the temperature-induced band renormalization.
5. As suggested by Reviewer #2, we enhanced the connection between superconductor-to-failed insulator transition and the AHE part.
6. The sheet resistance per layer is further clarified in the revision. The superconductor-to-failed insulator transition with a critical resistance per layer close to quantum resistance of Cooper pairs R_Q is further confirmed in the new device #8.
7. We plotted the resistance as a function of $1/T$ under different gating voltages in both device #5 and device #8.
8. The two major impacts of proton intercalation on AV_3Sb_5 , introducing disorders and carrier density modulation, have been well distinguished in the revised version.
9. We re-calculated the AHC by transforming Hall resistivity into Hall conductivity at first, and then extracted the AHC by subtracting the normal Hall conductivity part.
10. We studied the gate-dependent AHE in another device #7 with thickness around 50 nm to confirm the linear scaling relation $\sigma_{xy}^A \propto 0.14 \sigma_{xx}$, Fig. 4 is also updated accordingly.
11. The concentration of magnetic impurities has been estimated. We further specified the physical origins of large AHC in AV_3Sb_5 family.
12. We plotted σ_{xy}^A versus σ_{xx} diagram for a variety of materials and made a detailed comparison between our data and some other data in previous studies.
13. The Hall resistivity is redefined as R_{yx} in the revised version, we also updated Fig. 2 and Fig. 3 accordingly.
14. The references have been updated in the revision.
15. To strengthen the study, we made a major revision of our manuscript including title, abstract and main text. Some phrases are further polished to increase the readability.

Reviewer #1 (Remarks to the Author):

Comment[A0]: The authors reported a detailed study on charge modulation in CsV_3Sb_5 flakes using a protonic gating. They found a superconducting-insulator transition and an interesting evolution of carrier density and AHE with the variation of gating voltage. They argued that the dramatic change of AHE with decreasing gating voltage is attributed to the extrinsic-to-intrinsic transition of AHE. This study is worthwhile but, as a theorist, I am not persuaded that this study will significantly advance our understanding exotic phenomena in kagome metals.

Reply[A0]: We thank Reviewer #1 for taking the time to review this work, those insightful comments will no doubt help us to strengthen the present study. We also appreciate the reviewer's comment by saying "This study is worthwhile".

In this work, by use of the protonic gate, we for the first time observed the superconductor-to-failed insulator transition (SIT) induced by charged protons in a kagome superconductor, featuring a large saturated sheet resistance per layer (four orders of magnitude greater than the quantum resistance of Cooper pairs). According to the finite-size scaling analysis, we extracted the related critical exponent, which is **distinct** from those of the known 2D theoretical models, such as the classical percolation model and quantum percolation model. It probably results from the multiple SC gaps of multi-orbital/band nature of the electronic structure. The observation of SIT indicates that the newly emergent kagome superconductor is a nice and unique platform to investigate the quantum phase transition involving the intrinsic superconductivity intertwined with chiral charge order and giant anomalous Hall effect (AHE). It would inspire further studies on this key issue of interdisciplinarity among strong correlated physics and topological phases of matter.

The AHE has played a vital role in understanding low-power dissipation quantum phenomena associated with Berry phase and the development of the field of topological quantum matter. There are usually three distinct mechanisms to cause AHE: intrinsic mechanism due to the Berry phase of Bloch electrons and extrinsic mechanisms (skew scattering and side jump) related to the impurity scattering. Recent experiments have showed that the kagome metals exhibit the giant AHE in the low-temperature CDW phase. However, several fundamental issues about the nature and origin of this giant AHE in kagome metals with multiple orbital/band nature of electronic structure remain elusive.

In our work, we **first** clarified both intrinsic and extrinsic contributions of Dirac bands, electronic bands near the G-point and the heavy-hole flat bands at the saddle points, and found that the extrinsic skew scattering of heavy-hole bands at the saddle point has the major contribution to the giant AHE. **Second**, there are generally three main scenarios (the Kondo theory, the resonant skew scattering, scattering of carriers in bands with nonzero Berry curvature) to produce the extrinsic skew scattering and the resultant AHE [Sec. IV B in Rev. Mod. Phys. 82, 1539 (2010)]. We carefully excluded the first two scenarios and concluded that the giant AHE in AV_3Sb_5 is **unambiguously** dominated by the intense extrinsic skew scattering of holes in the nearly flat bands of VHS1 at the saddle points by multiple impurities. **More importantly**, we realized the crossover from the extrinsic skew scattering contribution to the intrinsic Berry phase contribution accompanied with dramatically changing AHC over a broad range against the modulation of Fermi level around the CDW gap. **To the best of our knowledge, it is the first example of the crossover from extrinsic skew scattering contribution to the intrinsic Berry phase contribution tuned by an electric gate.**

As a result, we could for the first time quantitatively determine the magnitude of intrinsic anomalous Hall conductivity in the kagome materials AV_3Sb_5 , providing direct evidence of the nontrivial chiral charge order and also benchmark to test various theoretical models for the chiral charge order. Interestingly, we found the linear scaling relation $\sigma_{xy}^A \propto 0.14 \sigma_{xx}$ through protonic gating in CsV_3Sb_5 , which is fundamentally distinct from the quadratic one in previous works [Refs. [23, 24] in the revised version].

Thus, our findings first uncover the unusual SIT and realize the crossover from the extrinsic skew scattering contribution to the intrinsic contribution of AHE and further unveil its close correlation with the chiral charge order in the kagome materials AV_3Sb_5 . Meanwhile, our work also provides some useful guidance to create and manipulate topological correlated electronic states and establish the generic phase diagram among those exotic correlated electronic states in kagome materials AV_3Sb_5 under electric gate, high pressure and high magnetic field.

The manuscript needs to clarify the following points.

We will give a point-to-point reply to all the concerns raised by Reviewer #1 in the followings.

Comment [A1]: With increasing of disorder, SIT is theoretically expected. Why is it exotic here? In addition, the authors only show the data for the device 5 with a thickness of 21 nm. How about the thinner flakes? Does the SIT only occur for thin flakes below a critical thickness?

Reply[A1]: We thank the reviewer's useful comments about SIT. It is known that there are two distinct theoretical scenarios for superconductor-insulator transition (SIT) in two dimensional electronic systems (2D). The fermionic scenario lies in the fact that the disorders diminish the effective attractive interaction and destroy the SC into normal metallic state. Based on general scaling arguments, the metallic states are absent at zero temperature in noninteracting disordered 2D electron systems with negligible spin-orbit interaction, which is named Anderson localization. In fact, the issue of 2D localization in disordered metallic samples with electron-electron interaction is more complicated and is still not fully understood. The other is the bosonic scenario, in which the global quantum coherence of SC state is loss due to the disorders and then the Cooper pairs become localized [V. F. Gantmakher and V. T. Dolgoplov, Phys Usp 53, 1 (2010)]. It gives us a conventional wisdom that SIT is theoretically expected in a superconductor with increasing of disorder in 2D.

In reality, more fascinating quantum phenomena emerge beyond this conventional wisdom of SIT. For example, a large number of experiments on a variety of systems reveal the exotic intermediate regime, quantum or anomalous metal whose zero-temperature conductivity can be orders of magnitude larger than the Drude conductivity, thus cannot be understood on the basis of Fermi liquid or Drude theory [A. Kapitulnik et al, Rev. Mod. Phys. 91, 011002 (2019)]. Meanwhile, on the insulating side, an initial resistivity divergence trend for $T \rightarrow 0$ K for a true insulator is replaced by a distinct resistivity saturation, exhibiting resistivity values that can be several orders of magnitude larger than the quantum resistance of Cooper pairs, dubbed as a "failed insulator".

Thus, for a new kind of complex electronic system, there usually exist several fundamental issues about SIT that are largely unexpected. First, whether the intermediate anomalous metal regime

exists or not? Second, whether the resistivity exhibits a saturation trend or divergent for $T \rightarrow 0$ K. Third, does it belong to the fermionic or bosonic scenario? Fourth, what is the universality class and the corresponding critical exponents? In addition, what is the thickness dependence of the critical behavior of SIT?

The layered kagome superconductor CsV_3Sb_5 is very unique due to the multiple orbital/band nature of electronic structure supporting a variety of novel quantum phenomena. First, the superconductor CsV_3Sb_5 was reported to support the robust zero-bias conductance peak inside the superconducting core through the STM spectroscopy, resembling the Majorana bound states in the superconducting topological insulator heterostructure. Second, under high pressure, there exists two superconducting domes in CsV_3Sb_5 , indicating the unconventional nature of superconducting phase. Third, the intrinsic superconductivity is intertwined with the chiral CDW order, nematic order and giant anomalous Hall effect.

Let us brief the key findings on SIT in the family of AV_3Sb_5 . Beside the field induced quantum metal regime in KV_3Sb_5 recently reported at ultra-low temperature, our present experiments revealed a saturated resistance trend for the temperature $T \rightarrow 0$ K in metallic state of SIT, and this saturated resistance trend survives even in the “insulating” state with a sheet resistance per layer four orders of magnitude greater than quantum resistance of Cooper pairs $R_Q \approx 6.45$ k Ω , that is the “failed insulator”. This “saturated resistance” for $T \rightarrow 0$ K is quite unusual in such a broad range (from $\sim 0.1R_Q$ to $\sim 10^4R_Q$), probably due to the incoherent tunneling between localized Cooper pairs. Second, finite-size scaling gives rise to distinct critical exponent from those of the conventional 2D theoretical models, indicating a new universality class.

With respect to the thickness dependence of SIT in CsV_3Sb_5 thin films, the SIT usually occurs in thinner samples with increasing disorders via proton intercalation. In thicker samples, however, the required density of disorders to induce SIT would be far beyond the capability of our protonic gate. Thus, it is expected that SIT only occur in thin flakes below a critical thickness. In order to clarify this point, we fabricated many more samples with different thicknesses. In thicker nanoflakes above 25 nm, we didn't observe SIT in our gating voltage range and the sheet resistance at 5 K is nearly unchanged except the change of carrier density. In those gating experiments, we found SIT mainly happened in thin nanoflakes at below 25 nm. It is worth emphasizing that gate-induced SIT in few layer CsV_3Sb_5 is technically challenging, this is because few layer CsV_3Sb_5 can be easily damaged during the proton intercalation before the sheet resistance per layer approaches R_Q , as we can see Fig. S3-2 (from 7 nm to 14 nm). However, we are able to repeat the gate-induced SIT in another sample (#8) with thickness ~ 18 nm, as shown in Fig. S3-3 in Supplementary Information. According to those experiments, we conclude that gate-induced SIT mainly occurs in thinner CsV_3Sb_5 nanoflakes with a critical thickness around 25 nm.

Revision: Following Reviewer's comments, we discussed the SIT in another thinner sample (#8) in Supplemental Materials (Fig. S3-3). We also presented the “failed insulator” in Fig. 1 of main text and updated the manuscript accordingly. The corresponding part now reads:

“Here, we show that electrically controlled proton intercalation has significant impacts on striking quantum phenomena in CsV_3Sb_5 nanodevices mainly through inducing disorders in thinner nanoflakes and carrier density modulation in thicker ones. Specifically, in disordered thin nanoflakes (below 25 nm), we achieve a quantum phase transition from a superconductor to a “failed insulator” with a large saturated sheet resistance for $T \rightarrow 0$ K.” (The Abstract, Page 2 in main text)

“In spite of the big R_s reaching up to $10^6\Omega$ on insulating side, however, a saturated resistance trend appeared for $T \rightarrow 0 K$, as shown in Fig. 1d. Note that this insulating state with a saturated resistance for $T \rightarrow 0 K$ is not a typical insulator but a “failed insulator”, probably due to the incoherent tunneling between localized Cooper pairs. This type of superconductor to “failed insulator” transition has also been observed in another sample #8 in Supplementary Fig. S3 and mainly results from the enhanced effective disorders due to the intercalated protons in the thinner nanoflakes with higher gate voltages” (Second paragraph, Page 5 in main text.)

Comment[A2]: The AHE is usually defined as the Hall effect without an external magnetic field. But here the Hall resistance always vanishes at a zero magnetic field. Why is it an AHE in the present experiments?

Reply [A2]: We thank the reviewer’s comments.

Generally, the nonlinear Hall trace can either be attributed to either two-carrier transport or anomalous Hall effect (AHE). In kagome superconductor, the antisymmetric “S”-shape Hall resistance R_{xy} has been observed in AV_3Sb_5 (A=K, Rb, Cs). Recent high-pressure experiment has illustrated that this “S”-shape Hall resistance co-occurs with charge density wave [Phys. Rev. B 104, L041103 (2021)] and cannot be simply explained by the two-band model, as we can see in the following figure R1.

Fig. R1. Under a pressure around 1.42 GPa, the Hall resistivity of CsV_3Sb_5 exhibits an antisymmetric sideways “S” line shape in the low-field region (Left) and can be attributed to AHE. Another kink occurs at high field region around 3 T, this kink still survives at 2.02 GPa where AHE disappears (Right). The nonlinear Hall part at 2.02 GPa can be well fitted by the two-band model. From [Phys. Rev. B 104, L041103 (2021)].

Those systematic experiments including high pressure, transport and STM et al support the topologically nontrivial properties of this correlated electron system and confirm the emergence of AHE in AV_3Sb_5 [Sci. Adv. 6, eabb6003 (2020); Phys. Rev. B 104, L041103 (2021); Nat. Mater. 20, 1353 (2021)]. But the nature and origin of giant AHE and its relation to the possible chiral charge order remain unclear and under debate.

It is known that the AHE usually occurs in the ferromagnetic systems with spontaneous magnetization in the absence of external magnetic fields. In fact, a finite magnetic field could play dual roles in electromagnetic transport phenomena of topological materials with topological energy bands: normal Hall effect via the Lorentz force and AHE via the Zeeman effect that breaks the degeneracy of energy bands and yields a net nonzero Berry curvature of energy bands. The

Berry curvature produces an anomalous velocity of Bloch electrons that is responsible for the intrinsic AHE, for example, 3D non-magnetic Dirac semimetal ZrTe₅ for observation of AHE. In particular, this field-dependent AHE would disappear at a zero magnetic field, e. g. ZrTe₅. In the layered Kagome superconductor AV₃Sb₅ (A=K, Rb, Cs), the giant AHE appears in the low-temperature CDW phase, the magnetic field is supposed to assist the formation of chiral charge density order (loop current order or chiral flux phase) from electronic states near the M points, which breaks the time reversal symmetry in the energy bands and leads to a large intrinsic AHE. Note that, this intrinsic contribution is one order of magnitude smaller than the huge AHC observed in experiments. Thus, besides the intrinsic part, the extrinsic skew scattering of holes in the flat bands near the M point would dominate the large AHC.

In the present work, we find that the carrier density in CsV₃Sb₅ can be largely tuned via the protonic gate, which helps us to discern the physical origins of this AHE at low magnetic fields. According to our gating experiments, we found that the giant AHE strongly depends on the position of Fermi level within CDW gap. Once the Fermi level is moved out of CDW gap, the AHE disappears. While a large AHE emerges when the Fermi level touches the upper sub-band of CDW gap with a large density of states (DOS). Moreover, we found an intrinsic AHE component independent of the longitudinal conductivity σ_{xx} when the Fermi level locates at the middle of the CDW gap where the DOS has been largely suppressed. The scaling analysis in different samples strongly indicates that the observed “S”-shape Hall component is AHE caused by the large skew scattering of the holes in the flat bands near the M points.

Comment[A3]: As the kagome metals CsV₃Sb₅ is a multi-orbital system, containing both holes and electrons, the carriers can exhibit complicated temperature dependent behavior. Therefore, a sign change of Hall resistance may not denote a Lifshitz transition. Using the protonic gating, the introduced disorder effect can also affect the carrier mobility. If the authors consider both electron and hole carriers, how will this affect the analysis for the evolution of Hall resistance?

Reply[A3]: We agree with Reviewer #1 that a sign reversal of Hall resistance during the change of the temperature may not be an unambiguous signature of a Lifshitz transition, since the Lifshitz transition in complicated multi-orbital systems can hardly be determined solely via transport properties. We note that a recent ARPES experiment revealed a temperature-induced band renormalization in RbV₃Sb₅ [PRX 11, 041010 (2021)], the sign change in temperature-dependent Hall resistance observed in CsV₃Sb₅ can also be attributed to this band renormalization due to the similar band structures as RbV₃Sb₅.

It has been shown that, in thinner samples, the proton intercalation can indeed largely induce disorders and simultaneously affect the carrier mobility, leading to superconductor-insulator transitions. In order to uncover the role of intercalated protons in the carrier density, we adopt two ways to minimize the impact of carrier mobility or disorders during the proton intercalation. On one hand, we choose much thicker nanoflakes to minimize the impacts of disorders that controls the low-temperature mobility (around 50-80 nm in thicker devices #4 and #7). On the other hand, we also apply much lower gating voltages (lower than 7V) to keep a lower proton concentration. As a result, the impact of proton gating on the carrier mobility could be limited in thicker samples.

We mainly focus on the nature and physical origin of giant AHE at low field region which is mainly contributed by the holes at the M point. We note that the transport below 10 T is dominated by

the holes at the M point with a linear Hall resistivity at low temperature region after the opening of CDW gap (except the AHE component at low field region), as shown in Supplementary Fig. S5. Thus, in our analysis, we roughly use a simple band model to extract the hole carrier density at the M points. Based on the gate-dependent carrier density near the M points, we deduced the giant AHE comes from the extrinsic skew scattering of the holes in the flat bands at the M points.

Revision: We added several sentences to clarify these points in main text, they now read:

“For each gating voltage, the temperature-dependent Hall effects demonstrate a sign reversal at the critical temperature T^ , probably due to the temperature-induced band renormalization”.*
(First paragraph, Page 8)

“Note that CsV_3Sb_5 is a multi-band kagome metal with its transport properties mainly determined by the hole pocket near the M points. Thus we can use a simple band model to fit this linear Hall resistivity at high field region and extract the approximate hole carrier density at the M-points.”
(First paragraph, page 8 in main text)

Comment[A4]: Fig. 2 only shows the data for thicker flakes (about 80 nm). Will similar behavior occur in thinner and thicker flakes? Is the sign reversal in Hall resistance a universal feature in kagome nano flakes?

Reply[A4]: We thank the reviewer’s useful comments. Actually, we tested more than 10 samples and indeed found that the similar behavior in nearly all of those gated samples. For example, the similar behavior happens in sample #3 (~95 nm) of Fig. S7 in Supplementary Information. Meanwhile, in thinner sample #5 (Fig. S3-1 in Supplementary Information), the negative voltages will also induce a sign reversal in Hall resistance, similar to that in Fig. 2 of the main text.

According to our gating experiments, we found the temperature-induced sign reversal of the Hall resistance only happens in hole band, probably due to the temperature-induced band renormalization [PRX 11, 041010 (2021)]. For the gating experiments at low temperatures (~5 K), we found that the proton intercalation can largely change the carrier density of the system and accordingly shift the Fermi surface from a hole pocket to an electron pocket. During this process, gate voltage will reverse the Hall resistance in both thicker (e.g. device #3, #4 and #7) and thinner (e.g. device #5 and #8) CsV_3Sb_5 samples. Thus, we think that the gate-induced sign reversal in Hall resistance is universal.

Revision: We added one sentence in main text to clarify the reviewer’s concern, it now reads

“This doping-induced sign reversal of Hall resistance has also been observed in other samples (Supplementary Fig. S7).”(First paragraph, page 8 of main text)

Comment[A5]: In principle the gating method should introduce a continuous charge tuning. Why is the carrier density in Fig.3c discontinuous?

Reply[A5]: We agree with Reviewer #1 that gating usually leads to a continuous (or a linear) charge tuning, because in most of the materials the DOS *continuously* decreases or increases with the shift of the Fermi level. This is consistent with our previous study in some other nano devices [Nano Lett. 21, 5599-5605 (2021); Nat. Commun. 12, 3639 (2021)]. In CsV_3Sb_5 , however, we repeated several

samples and found that the change of carrier density under protonic gate is indeed discontinuous. This is probably due to the complex evolution of DOS from multiple energy bands near the Fermi level in CsV₃Sb₅, as shown in Fig. R2 below.

Fig. R2. Band structures and DOS with different doping levels in CsV₃Sb₅.

Specifically, the energy bands are sensitive to the carrier doping. The hole (electron) doping will significantly shift the VHS1 upward (downward) with respect to the Fermi level and the distribution of DOS also changes accordingly during the doping, as we can see in Fig. R2. In addition, the DOS changes dramatically near the Fermi level (e.g. $N_e=0$), leading to an irregular change of the carrier density during the sweeping of the gate voltages from +6 V to -6 V. This complex evolution of DOS during the doping process may account for the *discontinuous* charge tuning.

Revision: We further clarified this point in main text, it now reads “We note that the discontinuity of the carrier density under gate voltage probably stems from the complex evolution of density of states (DOS) during the proton intercalation (See Fig. 4b below)” (First paragraph, page 9 in main text)

Comment[A6]: The kagome metals have been experimentally shown to be nonmagnetic but time-reversal-symmetry broken (by μ SR measurements Ref. 60, 61). However, in the theoretical calculations, the authors used a ferromagnetic state to calculate the AHC with a relatively large magnetic moment 0.25 μ_B . The assumed ground state is not consistent with experiment data.

Reply[A6]: We thank the reviewer for pointing out this issue about the numerical simulation of AHC. In our manuscript, we carried out the numerical calculation of intrinsic AHC by using a ferromagnetic (FM) state to mimic the contribution from field induced magnetization of the spin of V atoms. As shown in the Fig. S10, the calculated AHC over a broad energy region, exhibits a maximum ($\sim 1500 \Omega^{-1}\text{cm}^{-1}$) in the hole band, one order smaller than the maximum experimental value. This suggests that the intrinsic contribution from the Berry curvature of single-particle energy bands should not dominate the giant AHE in the AV₃Sb₅ materials. Moreover, because of the tiny observed magnetic moments of V atoms [E. M. Kenney et al J. Phys.: Condens. Matter 33, 235801 (2021).], the realistic AHC from field induced magnetization should be much smaller than the observed intrinsic one ($\sim 1100 \Omega^{-1}\text{cm}^{-1}$). Thus, the large intrinsic AHC in Fig. 4a mainly comes from the possible chiral charge order.

Revision: We would like to clarify this point in the revised version, it now reads “The calculated

AHC due to the field induced magnetization of the spin of V atoms over a broad energy region, exhibits a maximum ($\sim 1500 \Omega^{-1} \text{cm}^{-1}$) in the hole band, one order smaller than the maximum experimental value. This suggests that the intrinsic contribution from the Berry curvature of single-particle energy bands should not dominate the giant AHE in experiments. Note that, because of the tiny observed magnetic moments of V atoms, the realistic intrinsic AHC from field-induced magnetization should be much smaller than the observed intrinsic AHC.” (First paragraph, page 11 in main text)

Reviewer #2 (Remarks to the Author):

Comment[B0]: In this manuscript, Zheng et al studied the SIT and the AHE in kagome metal CsV_3Sb_5 nanoflakes with various thicknesses by using a protonic gate. In a 21-nm thin flake device, they observed a direct SIT and a large suppression of CDW induced by gating, which were interpreted in terms of the phase fluctuation due to enhanced-disorder. Meanwhile, in thick devices, electric gating changes the carrier density and shifts the Fermi level position across the CDW gap and gives rise to giant AHE. By further providing first-principle calculations, the authors demonstrated the giant AHE is induced by the skew scattering mechanisms of holes in the flat bands with finite Berry curvature. This experiment stands in the continuation of efforts of the studies of the recently discovered kagome material AV_3Sb_5 that incorporates electron correlation effects, band topology and geometric frustration. Although there have already been relevant studies on the transport properties of AV_3Sb_5 , the underlying physics of the SIT and the AHE have not been well addressed. Therefore, the electrical controls of SIT and AHE by gating have an opportunity to provide noteworthy results and may deepen people’s understanding on these interesting topics. I appreciate the authors’ efforts in the manuscript. However, I find the current data does not support the conclusions. More reproducible experimental results are required to support the claims. Moreover, there may be a flaw in the AHE data analysis process, which makes the interpretation of the AHE questionable. In spite of these concerns, I still find experiments interesting. However, at the current stage, I cannot recommend this article for publication without significant revisions.

Reply [B0]: We firstly thank Reviewer #2 for his/her insightful comments and positive appraisal by saying “the electrical controls of SIT and AHE by gating have an opportunity to provide noteworthy results and may deepen people’s understanding on these interesting topics” and by saying “I appreciate the authors’ efforts in the manuscript”.

Please consider the following questions and comments:

We will give a point-to-point reply to all the concerns raised by Reviewer #2 as follows.

Comment[B1]: It seems the SIT and the AHE part are two independent stories performed in samples with thin and thick thickness respectively. The connection between the two parts is too weak. To enhance the readability, I suggest the authors discuss the relationship between these two parts, rather than simply piecing them together.

Reply [B1]: We thank reviewer's comments and useful suggestions. We agree with Reviewer #2 that the connection between SIT and AHE was not strong enough in the previous version. In this revised version, we made a major revision to our manuscript to enhance the connection between those two parts. In addition, we also carefully polished the manuscript to increase the readability.

Revision: We made a major revision of the manuscript including Abstract and main text, we also enhanced the connection between those two parts by saying

"Here, we show that electrically controlled proton intercalation has significant impacts on striking quantum phenomena in CsV₃Sb₅ nanodevices mainly through inducing disorders in thinner nanoflakes and carrier density modulation in thicker ones. Specifically, in disordered thin nanoflakes (below 25 nm), we achieve a quantum phase transition from a superconductor to a "failed insulator" with a large saturated sheet resistance for $T \rightarrow 0$ K. Meanwhile, the carrier density modulation in thicker nanoflakes shifts the Fermi level across the charge density wave (CDW) gap and gives rise to an extrinsic-intrinsic transition of AHE". (First paragraph, page 2)

"In this work, we find that electrically controlled proton intercalation exhibits crucial impacts on the superconducting state, CDW state and the associated AHE in CsV₃Sb₅ nanoflakes via disorders and carrier density modulation. In thinner nanoflakes (below 25 nm) with large gate voltages (e.g. with an amplitude above 15 V), the enhanced disorders from intercalated protons suppressed both CDW and superconducting phase coherence and gave rise to a SIT associated with the localized Cooper pairs, featuring a saturated sheet resistance reaching up to $10^6 \Omega$ for $T \rightarrow 0$, dubbed a "failed insulator". While in thicker CsV₃Sb₅ nanoflakes with much lower gate voltages (within 7 V), the superconducting transition instead retained with nearly unchanged sheet resistance in normal state at 5 K, indicating very limited impact of disorder. " (Second paragraph, page 3)

"For a given gate voltage, the thick samples would diminish the impact of disorder of intercalated protons, leaving a large tune of the carrier density. Let us concentrate on the significance of the carrier density modulation on the AHE in CDW phase." (Third paragraph, page 7)

Comment[B2]: Is the observed SIT reproducible? The authors only provide the data of one sample in the main text (device #5). The quantized resistance $h/4e^2$ at the transition point is regarded as an important signature of SIT. However, when looking at the "SIT" data of other samples in the supplementary, it is clear that the critical resistance is not $h/4e^2$. Even considering the different thickness, one cannot get the quantized value $h/4e^2$. Therefore, it is likely that the observed $RQ = h/4e^2$ in device #5 is accidental.

Reply [B2]: We thank Reviewer for his/her insightful comments. We would like to point out that the realization of gate-induced SIT is technically challenging in very thin samples around or below 10 nm. This is because the thinner samples are very fragile and easy to be damaged during the gating process, especially for a large gate voltage above 15 V. In Fig. S3-2 of Supplementary Information, we can see that the protonic gate can tune the sheet resistance of those three thinner samples. However, those samples were unfortunately damaged during the gating process before the sheet resistance reached to the critical resistance of SIT. Thus, it is actually a gate-induced superconductor-to-metal transition in those samples between 14-7 nm in Supplementary Information.

In order to reproduce the experimental observations, we fabricated many more samples with different thicknesses and successfully repeated the observation of SIT in sample #8 with thickness ~ 18 nm, as shown in Fig. S3-3. It is clear that if we convert the critical sheet resistance to the sheet resistance per layer, the value will be very close to the quantum resistance R_Q (Fig. S3-3), akin to that in device #5. Moreover, we observed a “saturated resistance” for $T \rightarrow 0$ K on insulating side in both of our devices (#5 and #8), dubbed as a “failed insulator”, probably due to the incoherent tunneling between localized Cooper pairs. Based on the above discussions, we think these experimental observations are indeed superconductor-to-failed insulator transition.

Revision: We repeated the superconductor-to-failed insulator transition in another sample (device #8 with thickness around 18 nm). We also added a section in Supplementary Information to specify this point. The corresponding part now reads:

Supplementary Figure S3-3. Superconductor-to-failed insulator transition in device #8 (18 nm) under various gating voltages. **a.** Gate-dependent sheet resistance. Akin to device #5, we observed a superconductor-to-failed insulator transition with a critical sheet resistance $R_c \approx 371 \Omega$, as shown in **b.** If we convert the critical sheet resistance to the sheet resistance per layer, it has a value of 7420Ω , close to the quantum resistance of Cooper pair (R_Q). **c** Shows the sheet resistance as the function of $1/T$ under different gate voltages. Akin to device #5, it exhibits a ‘failed insulator’ state with a large saturated resistance for $T \rightarrow 0$ K on insulating side. **d** Multiple sets of $R_s(T, B)$ curves can collapse onto a single function, akin to a 2D SIT. **e.** By extracting the exponent product vz and plotting $\ln T_0$ versus $\ln |n_s - n_c|$ curve, we get $vz = 1.68$, close to the value obtained in sample #5. (Supplementary section 3-3, page 7)

Comment[B3]: To convince general readers that the phase transition is universal, more data analysis is needed. What are the critical exponents of the two samples (12 and 14 nm devices) shown in the supplementary materials?

Reply [B3]: We thank for the reviewer’s useful suggestions. To verify the universality of the superconductor-insulator transition (SIT), we did analysis on three more samples and replotted the gate-dependent sheet resistance (R_S). Since they were damaged during the gating process due to the fragility of those thinner samples, we failed to deduce the critical exponents of those two samples, in which the maximum sheet resistance did not reach the critical resistance of SIT. Fortunately, we successfully reproduced the SIT in another device (#8 with thickness around 18 nm) and analyzed the data accordingly. The critical exponent deduced from sample #8 is 1.68, close to that in device #5. The analysis of the data in device #8 further confirmed that the critical exponent deduced in those CsV_3Sb_5 nanoflakes are distinct from those of 2D conventional models for SIT such as classical percolation model ($\nu_z=4/3$), quantum percolation model ($\nu_z=7/3$), probably resulting from the complexity of SC gaps and multiple impurities.

Revision: We replotted the gate-dependent sheet resistance per layer in Fig. S3-2 and further analyzed the data. We also discussed the SIT in device #8 in Supplementary Information (See the reply to comment [B2]). The revised part in Fig. S3-2 reads

Supplementary Figure S3-2. Temperature-dependent sheet resistance per layer in thinner CsV_3Sb_5 devices under various gating voltages. a-c Temperature dependent sheet resistance per layer under different gate voltages for 14-7 nm devices, respectively. The CDW transitions gradually disappear with the increasing intercalated protons. As discussed in main text, superconductor-to-failed insulator transition occurs at a critical sheet resistance R_c . If we convert the critical sheet resistance R_c to the sheet resistance per layer, we found that this critical sheet resistance per layer is very close to the quantum resistance R_Q in device #5 and #8. In these three samples range from 14-7 nm, however, the sheet resistance per layer didn’t reach to the quantum resistance R_Q due to the failure of those devices under high gate voltages, demonstrating a superconductor-to-metal transition. The dashed line is the guideline of the quantum resistance R_Q . (Supplementary section 3-2, page 6).

Comment[B4]: In previous studies of SIT, the sheet resistance R_S is defined as the resistance divided by area. In this case, the critical resistance at the superconductor to Bose insulator transition point

reaches $R_Q = h/4e^2$. However, in the current manuscript, R_S has a different definition. To get a quantized value, the number of layers is considered. This is more like a definition of resistivity rather than resistance. For the 2D SIT phenomenon, it is the 2D sheet resistance R_S rather than the R_Q divided by layer number that should be universal. The authors should clarify this issue. Moreover, the protonic gating ability and the induced transport responses vary a lot for different layer. It is inappropriate to directly divide R by layer number.

Reply [B4]: We thank for the reviewer's insightful comments about the sheet resistance. We agree with the reviewer that the sheet resistance should be universal and reached a quantum resistance

R_Q in previous SIT in 2D. Generally, $R = \frac{\rho L}{tW} = R_S \frac{L}{W}$, with ρ being the resistivity of the material, t being the sheet thickness, L being the length, W being the width. The sheet resistance is usually defined by $R_S = \rho/t$. This definition of sheet resistance has been used in quasi-2D or interface systems. In the updated version, we replotted the temperature-dependent sheet resistance under various gating voltages, as shown in Fig. 1. We found the superconductor-to-failed insulator transition occurs at a critical sheet resistance $R_c \approx 316 \Omega$ in device #5 with thickness around 21 nm. In the layered CsV_3Sb_5 nanoflakes, the nanoflakes have finite thicknesses (or layers). The large resistivity anisotropy between the directions perpendicular and parallel to the layers [Ying Xiang et al Nat. Commun. 12, 6727 (2021)] indicates the weak interlayer coupling, implying the use of the sheet resistance per layer could be appropriate to some extent. So we discreetly use the sheet resistance per layer via $R_{S/layer} = (\rho/l)$ in our data analysis. In fact, the "sheet resistance per layer" had been used to investigate the SIT in various layered materials, for example, in FeSe thin films [R. Schneider et al, Phys. Rev. Lett. 108, 257003 (2012)] and $\text{La}_{2-x}\text{Sr}_x\text{CuO}_4$ [Xiaoyan Shi et al, Nat. Phys. 10, 437 (2014)] and $\text{La}_{2-x}\text{Ba}_x\text{CuO}_4$ [Yangmu Li et al, Sci. Adv. 5, eaav7686 (2019)].

We now convert this critical sheet resistance R_c to the critical sheet resistance per layer, that is $R_{c/layer} = (\rho_c/(t/n)) = (\rho_c/l)$, with n being the number of layers, l being the thickness of each layer. The obtained critical sheet resistance per layer in device #5 is $R_{c/layer} \sim 7268 \Omega$, which is very close to the value of quantum resistance $R_Q \approx 6450 \Omega$. Note that, the conventional sheet resistance usually varies in samples with different thicknesses, which had been historically used to tune the SIT in Bi films [D. B. Haviland et al Phys. Rev. Lett. 62, 2180 (1989)]. However, the sheet resistance per layer here is instead independent on the total thickness, which may reflect the universal features of SIT therein. It can be seen that the sheet resistance per layer still has the same dimension as the conventional sheet resistance. We also would like to point out that the sheet resistance and the sheet resistance per layer yield the same universality class and critical exponents of SIT in our work.

Unlike some other ions (e.g. Li^+), the proton has very high mobility due to its small size and mass. As a result, the distribution of the protons is usually spatially uniform, especially in thinner samples with a large van der Waals (vdW) gap. To further ensure the uniform distribution of the protons in each layer in our experiments, we wait for a period of time after stabilizing the gate voltage at 250 K followed by a slow cooling-down. In the revision, we repeated the SIT in another sample #8 with thickness ~ 18 nm [as shown in Fig. S3-3 in Reply to Comment [B2] above] and get the almost same critical exponent, suggesting the appropriateness of our data analysis.

Revision: We replotted the temperature-dependent sheet resistance in Fig.1. The main text is also

updated accordingly, the corresponding part now reads:

“We plot the sheet resistance as a function of carrier density near SIT at temperatures between 2 K and 50 K and obtain a critical resistance $R_c \approx 316 \Omega$ with a critical carrier density $n_c \approx 7.91 \times 10^{17} \text{ m}^{-2}$, as shown in Fig. 1c. Converting this critical resistance R_c to the sheet resistance per layer, $R_{c/\text{layer}} = (\rho_c/l)$ with l being the thickness of a single layer, we get $R_{c/\text{layer}} = 7268 \Omega$, very close to the quantum resistance of Cooper pair $R_Q \sim 6450 \Omega$.” (Second paragraph, page 5).

Comment[B5]: In Fig. 1b, there is a R_S saturation at low temperatures. It is apparently different from a true insulating state that exhibits exponential diverging behaviors (see ref. 27). I suggest the authors plot the resistance as a function of $1/T$. In the community of SIT, people usually adopt this method to identify whether the phase transition is an SIT or an SMT. In addition, in AV3Sb5 , there have already been reports on the metallic state rather than an insulator state when SC is killed (ref. 38). The authors should be careful when claiming a Bosonic SIT.

Reply [B5]: We thank Reviewer #2 for the insightful suggestion. Following this suggestion, we plot the sheet resistance as the function of $1/T$. Interestingly, the sheet resistance indeed exhibits a situation at low temperature in a large range below or above SIT. Specifically, we find a saturated resistance trend for the temperature $T \rightarrow 0 \text{ K}$ in metallic state in both SIT samples. Moreover, this saturated resistance trend survives even in the “insulating” state with a sheet resistance per layer four orders of magnitude greater than quantum resistance of Cooper pairs $R_Q \approx 6.45 \text{ k}\Omega$. It should be pointed out that it is quite unusual to observe a saturated sheet resistance in such a broad range (from $\sim 0.1R_Q$ to $\sim 10^4R_Q$). Also, the observed huge sheet resistance per layer (above 10^4R_Q) for $T \rightarrow 0 \text{ K}$ cannot be a metallic state. The large saturated resistance on insulating side for $T \rightarrow 0 \text{ K}$ is a ubiquitous feature of “failed insulator” [Sci. Adv. 5: eaav7686 (2019)], which is probably due to the incoherent tunneling between localized Cooper pairs. From the temperature dependent sheet resistance under various gating voltages in device #5 and #8 [Fig. 1b in the main text and Fig. S3 in Supplemental Information], the feature of our experimental data resembles the typical behaviors of SIT driven by electric fields, as shown in Fig. R3.

Figure R3. Superconductor-insulator transition driven by electric fields in $\text{La}_{2-x}\text{Sr}_x\text{CuO}_4$ [left, Nature 472, 458 (2011)] and in $\text{YBa}_2\text{Cu}_3\text{O}_7$ [right, Phys. Rev. Lett. 107, 027001 (2011)].

We note that a possible quantum metallic state was reported in KV_3Sb_5 at ultra-low temperature (from 0.05 K to 0.4 K) when SC is killed by a low magnetic field [Ref. 38]. The quantum or anomalous

metal appears as an intermediate regime, which either undergoes a crossover to a normal metal phase or further enters into insulating phase. In our experiments, we did not find discernible evidence of the quantum metal phase in CsV₃Sb₅ flakes at relatively high temperatures (above 2 K).

Based on the present experimental results, we would like to claim the SIT rather than the SMT. In fact, we do not exclude the existence of intermediate quantum metal phase in CsV₃Sb₅ flakes at ultra-low temperatures, which definitely deserves further comprehensive study through various experimental techniques in future [such as Science 366, 1505 (2019)].

Revision: We plotted the sheet resistance as the function of 1/T in Fig. 1. We also updated the discussion of saturated resistance in main text. It now reads

“We plot the sheet resistance as a function of carrier density near SIT at temperatures between 2 K and 50 K and obtain a critical resistance $R_c \approx 316 \Omega$ with a critical carrier density $n_c \approx 7.91 \times 10^{17} m^{-2}$, as shown in Fig. 1c. Converting this critical resistance R_c to the sheet resistance per layer, $R_{c/layer} = (\rho_c/l)$ with l being the thickness of a single layer, we get $R_{c/layer} = 7268 \Omega$, very close to the quantum resistance of Cooper pair $R_Q \sim 6450 \Omega$. In spite of the big R_s reaching up to $10^6 \Omega$ on insulating side, however, a saturated resistance trend appeared for $T \rightarrow 0 K$, as shown in Fig. 1d. Note that this insulating state with a saturated resistance for $T \rightarrow 0 K$ is not a typical insulator but a “failed insulator”, probably due to the incoherent tunneling between localized Cooper pairs.” (Second paragraph, page 5).

Comment[B6]: How to demonstrate the direct SIT is induced by enhanced disorders? What kind of disorder plays a role here?

Reply [B6]: We thank the reviewer for insightful comment about the specific role of disorders on SIT. In our ungated samples with giant AHE, the corresponding longitudinal conductivity is about $6 \times 10^5 \Omega^{-1} cm^{-1}$, implying that the sample belongs to the high conductivity regime accompanied with a small amount of the disorders from the defects and magnetic impurities. When increasing the gate voltage, the protons diffuse into the sample and both longitudinal conductivity and AHC decrease due to the carrier modulation and disorders. Because of the multiple SC gaps associated with different Fermi surfaces, the SC pairing turns out to be insensitive to the carrier modulation during the proton intercalation. Further increasing the gate voltages, the protons grow in amount and would enhance the impact of the charged disorder. Accordingly, the quantum coherence of SC order parameter starts to lose gradually, then the SC phase transits into the localized Cooper pairs, which is “failed insulator” in the revised version. Thus, the charged disorders from the intercalated protons would play a key role in the transition from a superconductor to a failed insulator.

Revision: We would specify the role of charged disorders from the intercalated protons in the transition from superconductivity to failed insulator in the revised main text. It now reads

“In thinner nanoflakes (below 25 nm) with large gate voltages (e.g. with an amplitude above 15 V), the enhanced disorders from intercalated protons suppressed both CDW and superconducting phase coherence and gave rise to a SIT associated with the localized Cooper pairs, featuring a saturated sheet resistance reaching up to $10^6 \Omega$ for $T \rightarrow 0$, dubbed as a “failed insulator.” (Second paragraph, page 3 of main text).

“This type of superconductor to failed insulator transition has also been observed in another

sample #8 in Supplementary Fig. S3 and mainly results from the enhanced effective disorders due to the intercalated protons in the thinner nanoflakes with higher gate voltages” (First paragraph, page 6 of main text).

Comment[B7]: The authors stated that the protonic gate plays different roles in thin and thick flakes, which are significant for the SIT and AHE. An important question is how the authors could distinguish between intercalation and carrier density modulation simply based on the current transport experiments?

Reply [B7]: We recognize that the two primary impacts of intercalated protons on CsV₃Sb₅ nanoflakes of different thicknesses under the protonic gate, inducing disorder and carrier density modulation, were not well discussed and distinguished in the previous version. In principle, those two impacts coexist during the whole gate process in both thin and thick nanoflakes.

However, in thinner nanoflakes below 25 nm under much higher gate voltage (e.g. with amplitude larger than 15 V), we found that the sheet resistance changes dramatically under the gate voltages. This dramatic change of the sheet resistance appears in all those thinner samples exhibiting SIT or SMT, where the disorders could break the quantum coherence of SC phase and the Cooper pairs tend to be localized, leading to large sheet resistances in the insulating state. Meanwhile, the carrier densities in those thinner samples were also largely tuned (e.g. energy band changed from a hole band to an electron band), as shown in Fig. S3-1 in Supplementary Information.

In thick nanoflakes above 25 nm under much lower gate voltages (≤ 6 V), however, we found that the sheet resistance of those thicker samples did not change significantly, and almost all of RT curves exhibit a metallic behavior with a superconducting transition near 5 K, as shown in Supplementary Fig. S4. Thus, the impact of disorder on thicker samples is diminished and not sufficient to destruct the quantum coherence of SC phase. For the Hall effect in those thick samples, however, we found the Hall resistance changes dramatically, indicating a large tune of carrier density. Thus, the major impact of a lower gate voltage in thicker sample is the carrier density modulation. Consequently, the anomalous Hall effect exhibits a nontrivial evolution against the gate voltage, in addition to a suppression of longitudinal conductivity.

Revision: To address Reviewer’s concern, we revised the draft accordingly. The corresponding parts now read:

“we find that electrically controlled proton intercalation exhibit crucial impacts on the superconducting state, CDW state and the associated AHE in CsV₃Sb₅ nanoflakes via disorders and carrier density modulation. In thinner nanoflakes (below 25 nm) with large gate voltages (e.g. with an amplitude above 15 V), the enhanced disorders from intercalated protons suppressed both CDW and superconducting phase coherence and gave rise to a SIT associated with the localized Cooper pairs, featuring a saturated sheet resistance reaching up to $10^6 \Omega$ for $T \rightarrow 0$, dubbed a “failed insulator”. While in thicker CsV₃Sb₅ nanoflakes with much lower gate voltages (within 7 V), the superconducting transition instead retained with nearly unchanged sheet resistance in normal state at 5 K, indicating very limited impact of disorder.” (Second paragraph, page 3).

“For a given gate voltage, the thick samples would diminish the impact of disorder of intercalated protons, leaving a large tune of the carrier density. Let us concentrate on the significance of the carrier density modulation on the AHE in CDW phase. We choose thicker CsV₃Sb₅ nanoflakes (~100

nm) with much lower gate voltages within 7 V and find that the proton intercalation mainly affects the carrier density in those thicker ones, leading to only a slight modulation of SC transition temperature (Supplementary Fig. S4), which is consistent with recent reports” (Second paragraph, page 7).

Comment[B8]: An “S” -shaped R_{xy} was attributed to the field induced AHE (Fig. 2a). How to exclude the possibility that it is due to the coexistence of electron- and hole-type carriers?

Reply [B8]: We agree with Reviewer #2 that the “S”-shape R_{xy} may also be induced by two-type carrier transport. In general, the nonlinear Hall trace can be attributed to either the two-carrier transport or anomalous Hall effect (AHE). In kagome superconductor, the antisymmetric “S”-shape Hall resistance R_{xy} has been observed in AV_3Sb_5 (A=K, Rb, Cs). Previous high-pressure experiment illustrates that this “S”-shape Hall resistance co-occurs with charge density wave [Phys. Rev. B 104, L041103 (2021)] and cannot be simply explained by the two-band model, as shown in the following figures from [Phys. Rev. B 104, L041103 (2021)].

Under a pressure around 1.42 GPa, the Hall resistivity of CsV_3Sb_5 exhibits an antisymmetric sideways “S” line shape in the low-field region (Left), while another kink happens at high field region around 3 T, this kink still survives at 2.02 GPa where AHE disappears (Right). Also, the nonlinear Hall part at 2.02 GPa can be well fitted by the two-band model. Those observations indicate that the “S”-shape Hall trace at low field region cannot be explained by the two-band model and should be attributed to AHE.

Those systematic experiments including high pressure, transport and STM et al indicate the topologically nontrivial properties of this correlated electron system and support the emergence of AHE in AV_3Sb_5 [Sci. Adv. 6, eabb6003 (2020); Phys. Rev. B 104, L041103 (2021); Nat. Mater. 20, 1353 (2021)]. Moreover, the carrier density in CsV_3Sb_5 can be largely tuned via the protonic gate in our study which helps us to discern the physical origins of the giant AHE at low field. According to our gating experiments, we found that the AHE is closely correlated with the position of Fermi level within the CDW gap near the M points. Specifically, once the Fermi level is moved out of the CDW gap, AHE disappears, while a large AHE emerges when the Fermi level touches the upper sub-band with a large DOS. Moreover, we found that an intrinsic AHE component is independent of the longitudinal conductivity σ_{xx} when the Fermi level locates in the CDW gap where the DOS is largely suppressed. We also conducted the scaling analysis in different samples, those observations and analysis strongly indicate that the observed “S”-shape Hall component is the AHE caused by the large skew scattering of the holes in the flat band near the M points.

Comment[B9]: A sign reversal was observed in the temperature dependent measurements and was attributed to the temperature-induced Lifshitz transition. Is it possible that this sign reversal is induced by the variation of gating ability at high temperatures?

Reply [B9]: We thank Reviewer #2 for the nice comments. The sign reversal of the Hall resistance was widely observed in CsV₃Sb₅ nanoflakes, even in the absence of a protonic gate. It is consistent with our bulk measurements and other transport studies [Phys. Rev. B 104, L041103 (2021)]. Meanwhile, we recognize that the evidence for temperature-induced Lifshitz transition in AV₃Sb₅ with multiple bands from three distinct kinds of carriers cannot be determined solely by transport experiments. According to recent ARPES results, temperature-induced sign reversal of Hall effect might originate from the temperature-induced band renormalization [PRX 11, 041010 (2021)].

It is worth pointing out that gate voltage was applied at above 250 K in our experiments, once the temperature is cooled down to below 200 K, the protons will be stabilized in CsV₃Sb₅ nanoflakes. We agree with the reviewer that the gate ability at high temperature might be different each time. However, the transport properties of our nanodevices at low temperatures mainly depend on the carrier doping induced by proton intercalation. If we sweep the gate voltage back and forth, we will find that the Hall resistance is directly correlated to the carrier density, and the magnitude of the gate voltage is not directly relevant, as we can see in Fig. 2 of main text. So we think that that this sign reversal is mainly induced by the charge doping.

Revision: We also clarified this point in the updated version, it now reads:

“For each gating voltage, the temperature-dependent Hall effects demonstrate a sign reversal at the critical temperature T^ , probably due to the temperature-induced band renormalization”.*
(First paragraph, Page 8)

Comment[B10]: When discussing the gate-dependent AHE, the authors decomposed the total Hall resistivity ρ_{-xy} into two parts: the normal Hall effect ρ_{-Nxy} and the anomalous Hall part ρ_{-Axy} . To quantitatively compare the AHE in AV₃Sb₅ with that in other systems, one has to convert the resistivity ρ_{-Axy} into conductivity σ_{-Axy} first. However, the way that the authors got σ_{-Axy} is wrong. The correct procedure is to transform ρ_{-xy} into σ_{-xy} through the tensor relationship $\sigma_{xy} = \rho_{xy} / (\rho_{2xy} + \rho_{2-xx})$ first, and then extract the σ_{-Axy} by subtracting the normal Hall conductivity σ_{-Nxy} (see refs. 23 and 47). However, in the current work, the authors first got ρ_{-Axy} , and then get σ_{-Axy} using the formula $\sigma_{Axy} = \rho_{Axy} / (\rho_{Axy}^2 + \rho_{2-xx})$. This data analysis operation is invalid and may lead to serious problems. Because the giant AHE is one major selling point, the authors should carefully check the data analysis process and replot the AHC figures.

Reply [B10]: We thank Reviewer #2 for the careful reading of manuscript and pointing out this issue. In previous version, we found that the longitudinal resistivity ρ_{xx} is much larger than the Hall resistivity ($\rho_{xx} \gg \rho_{yx} = \rho_{yx}^N + \rho_{yx}^A$). In this situation, we approximately extracted the anomalous Hall conductivity via the formula like $\sigma_{xy}^A = \rho_{yx}^A / (\rho_{yx}^A{}^2 + \rho_{xx}^2)$. We agree with the reviewer that this method is not rigorous. Following the reviewer's suggestion, we replotted the

AHC figures and updated the main text according.

Revision: We replotted the AHC figure (Fig. 4a) and updated the main text. The corresponding part now reads:

“To get the AHC σ_{xy}^A , we first convert the Hall resistivity into the Hall conductivity $\sigma_{xy} = \rho_{yx}/(\rho_{yx}^2 + \rho_{xx}^2)$, followed by linearly fitting the conductivity at high field and subtracting the normal Hall conductivity σ_{xy}^N .” (Page 9 in main text)

Comment[B11]: In Fig. 4a, the authors summarized the relation between σ_{Axy} and σ_{xx} , and claimed “the AHE in device #4 can be well captured by a linear scaling relation”. From the figures, I cannot see any linearity between σ_{Axy} and σ_{xx} . In my opinion, it is unscientific to do linear fitting only based on three data points.

Reply [B11]: We thank Reviewer #2 for careful reading of our work and useful comments. To further clarify the scaling relation between AHC σ_{xy} and longitudinal conductivity σ_{xx} , we fabricated another thick sample (#7 around 50 nm) and retested the anomalous Hall effects in detail under different temperatures. As shown in Fig. 4 of main text, we confirm that the AHC can be well fitted by a linear scaling relation.

Revision: We added the data from device #7 to Fig. 4 to testify the linear scaling relationship between σ_{xy} and longitudinal conductivity σ_{xx} . The caption of Fig. 4a is also updated.

Comment[B12]: In the AHE community, double log-scale plots are usually adopted when σ_{Axy} and σ_{xx} span several orders of magnitude (see ref. 24). Obviously, the variations of σ_{Axy} and σ_{xx} are too small here. It is unnecessary and misleading to use double-log scales.

Reply [B12]: In the revision, we revised the Fig. 4a in main text, double log-scale plots have already been removed.

Comment[B13]: The giant AHE was attributed to the skew scattering of holes by impurities. Could the authors estimate the level of magnetic impurities in the samples? For a nonmagnetic SC material, it is hard to imagine an extrinsic mechanism could give rise to such a large AHC.

Reply [B13]: We thank the reviewer for the useful suggestion and comments about the detail of skew scattering mechanism for AHE in AV_3Sb_5 .

It has been shown that in the clean limit, the anomalous Hall conductivity is dominated by the extrinsic skew scattering contribution. The linear scaling relation between σ_{xy} and σ_{xx} in Figure 4 in the main text implies that the skew scattering contribution increases linearly in the Bloch state lifetime τ as the longitudinal conductivity. Both the skew scattering AHC σ_{xy} and σ_{xx} are suppressed by increasing the impurity scattering. The skewness factor $S = \sigma_{xy}/\sigma_{xx}$ is independent of σ_{xx} when the skew scattering mechanism dominates. From the transport experimental data in our CsV_3Sb_5 thin flakes, the skewness factor S is about 0.03.

Since there exist some debate about the nature of chiral CDW phase supporting large AHE and no simple and microscopic model for the chiral CDW phase, it may hinder us to reach the precise

determination of the level of magnetic impurities in samples. However, we would like to estimate the concentration of impurities in our CsV₃Sb₅ samples from some empirical formulas for the skewness factor $S \approx (E_{SO}/E_F)(h/2\pi \tau n_{imp}V_{imp})$, where τ is the relaxation time, h is the Planck's constant, the spin-orbital interaction energy E_{SO} refers the CDW gap for chiral CDW phase, E_F is the Fermi energy, n_{imp} is the impurity concentration and V_{imp} is the impurity potential strength [S. Onida et al Phys. Rev. B 77, 165103 (2008)].

From Figure 4 in our manuscript, in the high temperature normal state, the Fermi level lies above the flat band near the M point, in the low temperature CDW phase, the flat band gets split into two CDW bands, the Fermi level crosses the upper CDW band. It implies that the magnitude of CDW gap is comparable to the Fermi level. We further assume that the impurity potential strength is the order of eV and the relaxation time is about 1 ps. As a result, we could estimate the approximate concentration of impurities to be the order of 1%, which includes the nonmagnetic impurities and paramagnetic impurities. It should be noted that, the level of paramagnetic impurities (about 0.7%) has been extracted from the Curie-Weiss tail in the magnetic susceptibility of in bulk CsV₃Sb₅ [F. H. Yu et al Phys. Rev. B 104, L041103 (2021)]. In addition, the mechanical exfoliation of the CsV₃Sb₅ nanoflakes from the bulk may further introduce defects or impurities. It is worth noting that, the accurate determination of the level of nonmagnetic and/or magnetic impurities in CsV₃Sb₅ is crucial to the understanding the nature of giant AHE and deserves further studies.

Regarding the puzzle of the large AHE in a nonmagnetic SC material, we would like to note that the giant AHC was merely observed in the CDW phase or chiral charge density wave phase with transition temperature around 90 K. As the temperature approaches to the SC transition temperature at around 5 K, the sample enters into the SC phase. Meanwhile, to the best of our knowledge, there is no experimental evidence of large AHE in SC phase reported in previous works as well as our present experiments. Recently, μ SR studies in CsV₃Sb₅ have reported that time-reversal symmetry is not broken in the SC state [arXiv: 2108.01574]. In fact, the absence of large AHE in the SC phase in AV₃Sb₅ materials still lacks the microscopic theoretical understanding.

Revision: We added some brief discussion of the estimation of level of magnetic impurities in the Supplementary Information. It now reads:

"We now estimate the concentration of impurities in our CsV₃Sb₅ samples from some empirical formulas for the skewness factor $S \approx (E_{SO}/E_F)(h/2\pi \tau n_{imp}V_{imp})$, where τ is the relaxation time, h is the Planck's constant, the spin-orbital interaction energy E_{SO} refers the CDW gap for chiral CDW phase, E_F is the Fermi energy, n_{imp} is the impurity concentration and V_{imp} is the impurity potential strength. The approximate concentration of impurities to be the order of 1%, which includes the nonmagnetic impurities and paramagnetic impurities." (Second paragraph, page 15 in Supplementary Information)

Reviewer #3 (Remarks to the Author):

Comment[C1]: In manuscript "Electrically controlled superconductor-insulator transition and giant anomalous Hall effect in kagome metal CsV₃Sb₅ nanoflakes" by G. Zheng et al., they present the results of protonic gate control of CsV₃Sb₅ nanoflakes. They succeeded in observing the superconductor to insulator transition. In addition, they demonstrated the tuning of anomalous

Hall effect by the protonic gate. They showed many results and each data looks quite interesting.

Reply [C1]: We first thank Reviewer #3 for taking the time to review our paper. We also appreciate the reviewer for the positive appraisal by saying “They showed many results and each data looks quite interesting”.

Comment[C2]: However, it is difficult to understand the overall feature of this work from this manuscript. For example, Fig. 4a seems to be quite interesting result because the anomalous Hall effect was able to be controlled into Skew scattering regime by gate. However, they plot only their present data on this Figure. By adding other data points from previous reports such as Ref [24 (PRB 104)], we can find the importance of this work.

Reply [C2]: We thank the reviewer for the instructive suggestion on the presentation of AHE data. Following this suggestion, we made a detailed comparison between the data in this work and other data in previous studies.

Revision: We add a new section in Supplementary Information (section 8) to address this concern raised by Reviewer, the corresponding part now reads:

Supplementary Figure S8. σ_{xy}^A versus σ_{xx} for a variety of materials. There are three different regimes including bad-metal hopping, intrinsic and skew scattering spaced by vertical lines. As we can see, the data in this work locates at the same region with previous studied KV_3Sb_5 and CsV_3Sb_5 . However, different from previous studies where $\sigma_{xy}^A \propto \sigma_{xx}^2$, the maximal AHC obtained in proton intercalated devices exhibit a linear scaling relation $\sigma_{xy}^A \propto 0.14 \sigma_{xx}$. Data from ref. [2,7-16]. (Supplementary section 8, page 13)

Comment[C3]: As for the superconducting issues, I recommend to depict the phase diagram T_c , T_{CDW} , T^* vs gate voltage or carrier density. With such phase diagram, we can understand

smoothly what they achieved.

Reply [C3]: We thank the reviewer for nice suggestion about the depiction of the phase diagram, which indeed enhances the readability of our manuscript. We modified the discussion of the phase diagram T_c , T_{CDW} , T^* vs gate voltage or carrier density in the revision.

Revision: We added several sentences in main text to adjust the discussion of the phase diagram T_c , T_{CDW} , T^* vs gate voltage or carrier density. We also made a major revision of the draft to increase the readability. Some of them now read:

“Applying a protonic gate, SC is evidently suppressed and disappeared when $V_g \leq -2 V$ ” (Second paragraph, Page 5 in main text).

“CDW transition temperature $T_{CDW} = 85 K$ at $V_g = 0 V$ gradually decreases to $73 K$ at $V_g = -6.5 V$ where SC has totally been suppressed”, “The non-synchronous disappearance of SC and CDW reveals that SC is more sensitive to disorder.” (Second paragraph, Page 7 in main text).

Comment[C4]: Regarding the analysis of Hall effect, I'm confused about the labels: R_{xy} and ρ_{xy} . I'm sure the equation: $\sigma_{xy}^A = -\rho_{xy}^A / (\rho_{xy}^A + \rho_{xx}^2)$ in page. 8. However, Hall resistance and Hall resistivity should be R_{yx} and ρ_{yx} . So that, the slope of Hall effect become negative at $V_g = 4.5V$ and $5 K$, because the y-axis of Fig. 2a is R_{xy} , not R_H ($R_{yx} (= -R_{xy})$). If they define the y-axis of Fig. 2a is Hall resistance, the carrier type is correct as hole at $V_g=4.5V$ and $5 K$, but the sign of anomalous Hall effect become negative. These relationship is opposite in Fig. 3b. The authors should clarify this point. Thus, at the moment, I would not recommend the publication of this work in Nat. Commun.

Reply [C4]: We thank Reviewer #3 for the careful elaboration of the measurement of AHE and insightful comments. Starting with the resistivity tensor, we can relate the current densities J_x , J_y and electric fields E_x , E_y by the expressions:

$$E_x = \rho_{xx}J_x + \rho_{xy}J_y;$$

$$E_y = \rho_{yx}J_x + \rho_{yy}J_y.$$

Generally, the current is applied along x-axis, that means $J_y = 0$ and the longitudinal resistivity $\rho_{xx} \equiv E_x/J_x$, the Hall resistivity $\rho_{yx} \equiv E_y/J_x$.

The above expressions can also be rewritten as:

$$J_x = \sigma_{xx}E_x + \sigma_{xy}E_y;$$

$$J_y = \sigma_{yx}E_x + \sigma_{yy}E_y = 0.$$

Combining the above equations, we can get the Hall conductivity $\sigma_{xy} = \rho_{yx}/(\rho_{yx}^2 + \rho_{xx}^2)$. We recognize that the Hall resistance shown in Fig. 2a should be R_{yx} and the carrier type is hole-type (this is also consistent with previous reports). However, the Hall conductivity $\sigma_{xy} = \rho_{yx}/(\rho_{yx}^2 + \rho_{xx}^2)$ shown in Fig. 3b now should be negative at $V_g = 4.5 V$. The sign of Hall conductivity shown in Fig. 3b was indeed opposite. In the revised version, we carefully checked and unified the notations of the Hall conductivity and Hall resistance.

Revision: We revised the main text accordingly. The anomalous Hall conductivities in Fig. 3b and Fig. 3c have also been updated. (Page 22 in main text)

REVIEWERS' COMMENTS

Reviewer #1 (Remarks to the Author):

I appreciate the significant efforts that the authors have made in response my comments and concerns. They have properly addressed the points I previously raised. Although this paper is not focusing on the controversial nature of CDW, superconductivity and nematicity in these V-based Kagome metals, their finding of SIT and tunable AHE in the thin flakes with protonic gating is still interesting. This paper will stimulate further studies on the intriguing phenomena in these Kagome metals. The experimental data seems reproducible and the conclusions are reasonable. If the experimental analysis of SIT is solid (I am not familiar with experimental details in this part, this can be checked by other referees), I think this paper deserves publication in Nature Communications.

Reviewer #2 (Remarks to the Author):

The authors have made significant revisions to the manuscript. I support its publication in nature communications.